# Investigation of the effect and mechanisms of moxa smoke in the treatment of Influenza A Virus (IAV) infection

**Ting Cao, Wenchao Pan, Ziyao Liang, Jingyu Quan, Miaona Zhang, Huameng Li, Long Fan, Xuhua Yu** ⓘ *

State Key Laboratory of Traditional Chinese Medicine Syndrome, The Second Affiliated Hospital of Guangzhou University of Chinese Medicine/ Guangzhou University of Chinese Medicine, Guangzhou, Guangdong, China

* dryuxuhua@gzucm.edu.cn

## Abstract

Influenza, primarily caused by the Influenza A virus, is a highly contagious respiratory disease. While moxa burning is a traditional method used to reduce respiratory infections, most studies have focused on the components of moxa and air disinfection, often neglecting the pharmacological effects and mechanisms of moxa smoke. This study aimed to explore the antiviral and anti-inflammatory effects of moxa smoke in vivo, as well as the underlying mechanisms involved. Utilizing multiple databases, we identified 52 components of moxa smoke that target 384 proteins, with 92 of these potentially linked to protection against H1N1. Network analysis conducted using Cytoscape revealed 16 core targets, including PPARG and STAT3. We performed molecular docking to verify the stable binding affinities of core compounds with their corresponding targets. In vivo experiments demonstrated that moxa smoke significantly decreased the number of inflammatory cells in bronchoalveolar lavage fluid (BALF), lowered the levels of H1N1 nucleoprotein (H1N1NP), and reduced the mRNA expression of cytokines with chemokines in lung tissue, including Il-6, Il-1β, Tnf-α, Cxcl1, Cxcl2, Cxcl10 and Ccl2. These results suggest a reduction in lung inflammation in mice infected with the PR8 strain of the IAV. Western blot analysis indicated that moxa smoke upregulated PPARγ and reduced phosphorylated STAT3 levels. GW9662 inhibited the reduction of recruitment of inflammatory cells by moxa smoke, but didn't inhibit the reduction of viral load after moxa smoke treatment. A four-day treatment did not cause functional injury to the lungs, kidneys, or liver of H1N1-infected mice. However, after four weeks of exposure to moxa smoke, the mice exhibited changes in organ weight and pathological damage in the lungs and kidneys. In summary, Moxa smoke suppressed influenza virus-induced inflammatory cell infiltration by upregulating PPARγ, while simultaneously reducing viral load through PPARγ-independent mechanisms. Short-term exposure to moxa smoke did not cause

**Data availability statement:** All relevant data are within the manuscript and its Supporting Information files.

**Funding:** The study was supported by the Guangzhou Key Laboratory of Traditional Chinese Medicine for the Prevention and Treatment of Chronic Cough and Dyspnea (Grant Number 2023A03J0226), the double world-class and high-level university discipline collaborative innovation team project of Guangzhou University of Chinese Medicine (Grant Number 2021XK27), and investigation into the Optimal Dosage of Flu Dual-Relief Granules for the Treatment of Viral Pneumonia and the Underlying Mechanisms Involving Energy Metabolism Regulation in Macrophage Polarization (Grant Number 2022B1515230001).

**Competing interests:** The authors have declared that no competing interests exist.

**Abbreviations:** ANOVA, analysis of variance; ARDS, acute respiratory distress syndrome; BALF, bronchoalveolar lavage fluid; BC, betweenness centrality; BCA, bicinchoninic acid; BMDMs, bone marrow-derived macrophages; BP, biological processes; CC, cellular components; CC, closeness centrality; CXCL, C-X-C motif chemokine ligand; DC, degree centrality; EC, eigenvector centrality; ECL, enhanced chemiluminescence; ECM, extracellular matrix; ECM, extracellular matrix; FCS, fetal calf serum; FDA, Food and Drug Administration; GM-CSF, granulocyte macrophage colonystimulating factor; GO, Gene Ontology; H&E, hematoxylin and eosin; IAV, influenza A virus; IL, interleukin; IOD, integrated optical density; KEGG, Kyoto Encyclopedia of Genes and Genomes; LAC, local average connectivity; MEM, minimal essential Eagle's medium; MF, molecular functions; MLI, mean linear intercept; MMP, matrix metalloproteinase; MS, moxa smoke; NAI, neuraminidase inhibitors; NC, network centrality; NP, nucleoprotein; NS, normal saline; OMIM, Online Mendelian Inheritance in Man database; PBS, phosphate-buffered saline; PDB, Protein Data Bank; PFU, plaque-forming units; PPAR, peroxisome proliferator-activated receptor; PPI, the protein–protein interaction; PVDF, polyvinylidene fluoride; SDS–PAGE, sodium dodecyl sulfate–polyacrylamide gel electrophoresis; SEM, standard error of the means; STAT, signal transducer and activator of transcription; TBST, Tris-buffered saline-Tween 20; TNF-α, tumor necrosis factor-α.

significant impairment of pulmonary, hepatic or renal function; however, prolonged exposure may result in respiratory and renal dysfunction, potentially leading to more severe adverse effects.

## Introduction

The Influenza A virus (IAV), classified within the Orthomyxoviridae family, is characterized by significant genetic variability and can be transmitted via airborne particles, respiratory droplets, and direct contact, thereby affecting both human and animal populations. Since its first appearance in the early 20th century, the H1N1 strain has been implicated in multiple global pandemics. The influenza pandemic of 1918–1919, which is estimated to have resulted in as many as 50 million fatalities, surpassed the death toll of World War I. This pandemic led to a significant reduction in life expectancy across numerous nations during its occurrence [1–2]. During the 2024–2025 influenza season, Japan experienced a substantial influenza epidemic. From September 2, 2024, to January 26, 2025, the cumulative number of influenza cases in Japan is estimated to have reached approximately 9.523 million (https://www.niid.go.jp/niid/ja/). Influenza outbreaks underscore the significant impact and seriousness of annual epidemics on the global population, as well as the strain they place on the healthcare systems of various nations. This is particularly evident in seasonal epidemics that have had a pronounced effect on low- and middle-income countries [3].

IAV infection induces complex lung pathology characterized by acute-phase damage including direct viral injury to alveolar epithelial cells, cytokine storm-mediated inflammation with immune cell infiltration [4–5], alveolar-capillary barrier disruption causing pulmonary edema, and potential progression to acute respiratory distress syndrome (ARDS)6. Chronic manifestations include the development of pulmonary fibrosis, exacerbated by secondary bacterial infections [4,6]. Histopathological evidence demonstrates acute-phase alveolar wall thickening and hyaline membrane formation, followed by chronic fibrotic remodeling and destruction of alveolar structures [4].

Effective strategies for combating Influenza A encompass the administration of vaccines, which may be either inactivated or live attenuated, as well as the utilization of antiviral medications for both prophylactic and therapeutic purposes [7]. Influenza vaccines demonstrate an efficacy rate ranging from 70% to 90% in healthy young adults; however, this effectiveness diminishes in the elderly population, with protective effects typically lasting only a few months [7–8]. The antiviral medications approved by the U.S. Food and Drug Administration (FDA) for the treatment of influenza encompass neuraminidase inhibitors (NAIs), M2 ion channel inhibitors, and viral RNA polymerase inhibitors [9]. These therapeutic agents are required to be administered continuously during periods of elevated viral activity [4]. Advanced cases of influenza may lead to bacterial or fungal infections, necessitating symptomatic treatment with antibacterial or antifungal agents. There is an urgent demand for novel antiviral drugs that can combat influenza while alleviating associated lung inflammation.

Traditional Chinese medicine encompasses a diverse array of antiviral ingredients; however, most of these must be administered orally and are subject to metabolic effects. Artemisiae argyi has been reported to possess antiviral properties and is primarily found in temperate and subtropical regions of the Northern Hemisphere, including countries such as China, Japan, and South Korea [10]. Historically, people have burned Artemisiae argyi to prevent infectious diseases. Clinical studies indicate that moxibustion, a traditional therapy involving the burning of moxa (a form of Artemisiae argyi), is effective in treating influenza [11]. The main components of Artemisia include volatile oils, flavonoids, phenolic acids, and triterpenoids [10]. Additionally, the volatile oil extracted from Artemisiae argyi exhibits anti-inflammatory properties [12–13]. However, most research focuses on the components of wormwood itself rather than those of moxa tobacco in relation to antiviral and anti-inflammatory effects. To date, the antiviral and anti-inflammatory effects of moxa smoke have not been verified in vivo, and the underlying mechanisms remain unclear.

Network pharmacology shifts the paradigm of drug discovery from a "one target-one drug" framework to a "multi-target-multi-component" strategy. This transition is consistent with the holistic and systematic nature inherent in Chinese herbal extracts [14]. As an established methodology for studying complex herbal mixtures, network pharmacology enables comprehensive investigation of moxa smoke's therapeutic mechanisms by constructing and analyzing multi-component, multi-target interaction networks. Therefore, the primary objective of this research was to identify the antiviral and anti-inflammatory effects of moxa smoke, as well as the underlying mechanisms through network pharmacology. Following target identification, a detailed functional characterization was conducted, after which a pathway-gene interaction network was developed to elucidate the key biological pathways influenced by the predicted targets of moxa smoke and their potential antiviral mechanisms against H1N1. Subsequently, molecular docking simulations were performed to assess the binding affinities and interactions between the bioactive constituents of moxa smoke and the core target proteins. To confirm therapeutic efficacy and gain mechanistic insights, in vivo validation was carried out using a murine model infected with the influenza virus, where pharmacodynamic evaluations were performed to substantiate the hypothesized modes of action.

## Materials and methods

### Network pharmacology

**Obtaining targets associated with MS and H1N1.** Moxa smoke (MS), a gaseous byproduct similar to cigarette smoke, contains numerous bioactive constituents. To compile its chemical profile, relevant literature was sourced from the China National Knowledge Infrastructure (CNKI), PubMed, and Google Scholar. After excluding toxic gaseous compounds, the pharmacologically active components of MS were identified based on their relative concentrations [15–19]. The PubChem database (https://pubchem.ncbi.nlm.nih.gov/) was used to obtain the SMLLE name of active components, which were then entered into the SwissTargetPrediction database (http://swisstargetprediction.ch/) to predict potential targets associated with MS, retaining only those with a probability >0.

For H1N1-related genes, we queried OMIM, GeneCards, and DisGeNET using search terms such as "influenza A virus", "influenza A" and "H1N1" with the species to "*Homo sapiens*". Concurrently, MS-related targets from previous studies. By integrating these datasets, we created a comprehensive list of disease-associated targets. Intersection analysis was conducted via the bioinformatics platform (http://www.bioinformatics.com.cn/) to pinpoint shared targets between MS's bioactive components and H1N1 pathogenesis.

**Network construction and network topological features.** To map the intricate interplay between moxa smoke (MS), H1N1, and their associated targets, we created an interaction network using Cytoscape 3.9.1. The overlapping targets were analyzed in the STRING database (https://string-db.org/) under species-specific filters (*Homo sapiens*) to generate a protein-protein interaction (PPI) network. The resulting network was subsequently imported into Cytoscape 3.9.1 for visualization. Hub gene selection was conducted by calculating degree centrality, identifying genes with values ≥2-fold the median as crucial in the treatment of H1N1 mediated by MS. Further refinement was achieved through

topological analysis using the CytoNCA plugin, which assessed nodes based on several metrics: degree centrality (DC), betweenness centrality (BC), closeness centrality (CC), eigenvector centrality (EC), network centrality (NC), and local average connectivity (LAC) [20]. Genes that scored above the median across these parameters were classified as core genes. For better visualization, the central network was displayed using the Network Analyzer plugin, where node prominence reflected degree centrality values.

**Enrichment analysis.** To elucidate the biological roles of the identified MS and H1N1-associated targets, we conducted Gene Ontology (GO) and Kyoto Encyclopedia of Genes and Genomes (KEGG) pathway enrichment analyses using the Metascape platform (https://metascape.org/). The analysis was configured with "OFFICIAL_GENE_SYMBOL" as the identifier type and was restricted to "*Homo sapiens*". The enrichment results were visualized using statistical plots that represent cellular components (CC), molecular functions (MF), biological processes (BP), and KEGG pathways. To enhance interpretability, gene-related elements in the pathway diagrams were color-coded, facilitating subsequent analysis and prediction of MS's potential therapeutic mechanisms against H1N1 infection.

**Molecular docking.** The three-dimensional structures of target proteins were retrieved from the RCSB Protein Data Bank (PDB) (https://www.rcsb.org/), while the molecular structures of bioactive components were obtained from PubChem (https://pubchem.ncbi.nlm.nih.gov/). Using PyMOL 3.7.9 and AutoDockTools 1.5.7, the protein structures were prepared by removing extraneous ligands and water molecules, isolating protein chains, incorporating nonpolar hydrogen atoms, and calculating Gasteiger partial charges. For docking simulations, the prepared proteins served as receptors, while the compounds acted as ligands. Both receptor and ligand structures were converted to PDBQT format using AutoDockTools 1.5.7. The molecular docking calculations were then performed with AutoDock Vina 1.2.5 to investigate potential binding interactions between the protein targets and bioactive compounds. Binding affinity was assessed based on calculated interaction energies, with more negative values indicating greater stability of the receptor-ligand complex. The most favorable docking conformation, representing the strongest predicted binding interaction, was selected and visualized using PyMOL 3.7.9 for structural analysis.

## Animals

Specific pathogen-free male or female BALB/c mice (age: 6–8 weeks; weight: 18–20 g) were purchased from Beijing Vital River Laboratory Animal Technology Co., Ltd. (Beijing, China). The experiments were conducted at Guangdong Huawei Testing Co., LTD's experimental center and Guangzhou University of Chinese Medicine (Guangzhou, China). The research complied with the guidelines set forth by the National Health and Medical Research Council of China and received ethical approval from the Animal Experiment Center of Guangzhou University of Chinese Medicine (Approval number: 20231025001; Application date: October 25, 2023).

## Moxa smoke exposure and drug treatment

Mice were exposed to Moxa smoke from moxa sticks (Chi Fang Laoren brand, manufactured in Qichun County, Hubei Province, China) of different weights. Simply, the raw material for making the sticks is three-year-old Qichun Artemisia argyi grown in Qichun County, Hubei Province, China. The leaves are sun-dried firstly and repeatedly ground to remove coarse and hard impurities, until one kilogram of fine Artemisia argyi floss is obtained from every ten kilograms of dried Artemisia argyi leaves. Then the floss is rolled into moxa sticks of 18 mm x 200 mm.

In this study, two concentrations of moxa smoke were utilized: low moxa smoke (LMS, 0.3 g moxa stick every 30 min, resulting in a PM10 concentration of approximately 1.335 mg/m³) and high moxa smoke (HMS, 0.6 g moxa stick every 30 min, resulting in a PM10 concentration of approximately 2.67 mg/m³) [21]. Both PM10 concentrations were lower than the typical concentration of 3.54 mg/m³ found in moxibustion clinics [22]. Different groups of mice were given corresponding concentrations of moxa smoke every day for four days or four weeks.

The exposure was conducted in the ventilation hood. The burned moxa sticks and the mice were placed in a 36-liter plastic chamber (60 x 40 x 15 cm) which is with 3 holes (5 mm of the diameter) on the lid to ensure air circulation. Mice were exposed to tobacco smoke three times daily at 9:00 AM, 12:00 PM, and 3:00 PM, with each session lasting 1.5 hours. After every 25 minutes of exposure, the chamber lid was fully opened for 5 minutes to ensure adequate air exchange within the chamber.

Control animals were housed in identical chambers without exposure to moxa smoke. Body weight measurements were recorded at the same time each morning before the initiation of treatment.

The PPARγ agonist Pioglitazone (PGZ; MCE, USA) was administered via oral gavage at a dosage of 60 mg/kg/day [23], commencing 4 day prior to PR8 nasal instillation and continuing for a total of 9 days until the animals were euthanized. Conversely, the PPARγ antagonist GW9662 (MCE, USA) was delivered by intraperitoneal injection at a dose of 2 mg/kg/day [24], starting at 8:00 am and one day before smoke inhalation exposure and maintained for ten days until the animals were sacrificed.

## H1N1(A/PR/8/34) virus infection

The experimental scheme is based on findings from previous research [25]. Following the four-day MS or sham exposure protocol, the animals were anesthetized with isoflurane on day 5 for viral challenge. Each mouse received 12 plaque-forming units (PFU) of H1N1 (A/PR/8/34) in 30 µL of Minimum Essential Medium (MEM) via intranasal instillation, utilizing a well-established infection model. Control animals were administered an equivalent volume of sterile MEM alone. All subjects were euthanized at the 5-day post-infection time point for subsequent analysis.

## Bronchoalveolar lavage fluid and lung collection

At the study endpoint (day 10), the mice were humanely euthanized using an overdose of sodium pentobarbital. The respiratory tract was then surgically exposed, and the trachea was cannulated for bronchoalveolar lavage (BAL) collection. The initial lavage was performed with 400 µL of phosphate-buffered saline (PBS) supplemented with 2% fetal calf serum (FCS), followed by three sequential washes of 300 µL each using FACS buffer. Following the collection of bronchoalveolar lavage fluid (BALF), the counting of nucleated cells was performed using a fluorescence microscope after the cells were stained with AO/EB. Subsequently, cytospins were prepared, and the cells were fixed and stained with Epredia Shandon Kwik-Diff Stains (Richard-Allan Scientific, Runcorn, England, United Kingdom). Then the cell differentials were assessed based on their morphological characteristics [25–26]. The cell count results were calculated using the following formula:

Final absolute cell count = Total number of cells × Percentage of each cell type Cardiac perfusion was performed through the right ventricle using 5 mL normal saline to achieve complete blood removal from pulmonary tissue. The lung tissues were removed as a whole, quickly washed with normal saline, dried with a blotting paper, and then promptly frozen in liquid nitrogen for storage at −80°C until they could be analyzed further.

## Histology

Following the airway lavage procedure, the upper lobe of the left lung was surgically excised from each murine subject and subsequently fixed in 4% paraformaldehyde for a duration exceeding 24 hours. The fixed lung tissues were then embedded in paraffin wax. Utilizing a microtome, sections with a thickness of 4 µm were prepared, affixed to glass slides, deparaffinized, and stained with hematoxylin and eosin (H&E) and Masson's trichrome to assess the extent of pulmonary inflammation and fibrosis. The stained sections were analyzed using a Nikon TI2-E microscope, and images at the tissue scale were captured. These images were processed and integrated using Nis-Elements software. Inflammation scores were assessed according to the methodology established by Curtis et al [27]. Each lung section was segmented into 9 areas for individual scoring, and the overall inflammation score for the lung section was calculated as the average of these scores.

Masson's trichrome staining was utilized to evaluate collagen deposition within the airways and lung parenchyma. The collagen-stained areas (blue) on each paraffin section were delineated microscopically, and quantification of the collagen-positive areas and integrated optical density (IOD) was performed using Image-Pro Plus 6.0 software (Version X; Adobe, San Jose, CA). These values were normalized to the total examined area and expressed as the percentage of collagen fibers.

Additionally, collagen content in lung tissue was quantified by measuring hydroxyproline levels. Lung samples were processed using a hydroxyproline assay kit (Nanjing Jiancheng Bioengineering Institute, Nanjing, China) according to the manufacturer's protocol. Briefly, lung tissues were weighed, homogenized in sterile water, and hydrolyzed in 6N hydrochloric acid at 100°C for 5 hours. The hydrolyzed samples were then incubated with 4-(Dimethylamino)benzaldehyde (DMAB) at 60°C for 15 minutes. The absorbance of the oxidized hydroxyproline was measured spectrophotometrically at 550 nm to determine collagen content.

Five regions of lung tissue pathological sections—specifically the upper, middle, lower, left, and right areas—were randomly selected, excluding the trachea and blood vessels. These regions were examined at 200 × magnification using an Nikon TI2-E microscope, and images were acquired with a digital camera. Cross lines were drawn at the center of each field of view, and alveolar spaces (Ns) was counted. The total length of the lines (L) was measured. The mean linear intercept (MLI), representing the average alveolar diameter, was calculated using the formula: MLI = L/Ns. Image analysis was conducted utilizing Image Pro-Plus 6.0 [28] software.

## Total RNA extraction and TaqMan quantitative PCR

Total RNA was extracted from entire lung tissues utilizing the RNeasy Mini Kit (Qiagen, Germany) in conjunction with β-mercaptoethanol, adhering to the protocols outlined by the manufacturer. Subsequently, 30 μL of RNA was reverse-transcribed into complementary DNA employing the RT Buffer Mix and RT Enzyme Mix (Thermo Fisher Scientific, USA). TaqMan qPCR analysis was conducted using the ABI QuantStudio™ 7 Flex (ABI, USA) and Applied Biosystems™ TaqMan® Fast Advanced Master Mix (Thermo Fisher Scientific, USA). GAPDH served as the internal reference gene. The Ct value, defined as the amplification cycle (out of 40 total cycles) where reporter fluorescence crosses the preset threshold, indicates target sequence detection. In this study, Ct values were converted utilizing the threshold cycle time (Ct) alongside the relative quantification method, and were subsequently expressed in relation to the levels of GAPDH mRNA.

## Western blot analysis

For protein isolation, 40 mg of pulmonary tissue was homogenized in 500 μL of RIPA buffer using 1 mL syringe. After 30 minutes of complete lysis on ice, the samples were centrifuged at 12,000 × rpm for 5 minutes at 4°C, and the protein-containing supernatant was carefully collected. Protein quantification was performed using a BCA assay in accordance with the manufacturer's protocol (Pierce Biotechnology). The protein samples were denatured by heating at 100 °C for 5 min after mixing with 4 × loading buffer. Equal protein quantities were electrophoresed on a 10% SDS-PAGE gel (Epizyme Biotech, China) and electroblotted onto polyvinylidene fluoride (PVDF) membranes (Merck, Germany).

The membranes were blocked for one hour at room temperature with a 5% non-fat milk solution and subsequently incubated overnight at 4 °C with primary antibodies targeting GAPDH (2118, 1:1000), p-STAT3 (9145, 1:2000), STAT3 (9139, 1:1000), and PPARγ (2430, 1:1000), all sourced from Cell Signaling Technology (CST).Following this incubation, the membranes were exposed to the corresponding secondary antibodies for one hour. Protein bands were detected by enhanced chemiluminescence (ECL; Millipore, USA) and imaged using the eBlot Touch chemiluminescence detection system (China). Image analysis and quantification were performed using ImageJ software package (National Institutes of Health, USA).

## Statistical analysis

The aggregated data are presented as the mean ± standard error of the mean (S.E.M.). The variable "n" represents the number of mice in each treatment group. Statistical significance was evaluated using one-way analysis of variance (ANOVA). All statistical evaluations were performed utilizing GraphPad Prism 9.3.0. A p-value of less than 0.05 was considered to indicate statistical significance.

## Result

### Network pharmacology

**Screening of targets of moxa smoke compounds and H1N1 infection.** To examine the association between moxa smoke and H1N1 infection, network pharmacology was conducted. According to the papers from existing studies, we conclude the components in Moxa smoke (Table 1). The potential targets of moxa smoke were identified through an analysis of relevant literature and the SwissTarget Prediction databases. Following the exclusion of targets with a probability score below zero, as determined by SwissTarget Prediction, and the removal of duplicates, a total of 384 targets were identified. In order to predict the pertinent targets associated with H1N1 infection, we conducted a screening of 3,409 genes from the aforementioned databases.

**The PPI network.** The intersection of moxa smoke and H1N1 infection targets is depicted in a Venn diagram (Fig 1A). Following this, the 62 common targets were entered into STRING version 12.0 to construct a protein-protein interaction (PPI) network, with a minimum interaction score set at 0.400 (Fig 1B). The primary targets of MS involved in its anti-H1N1 infection effect are illustrated in Table 1, which comprises 16 proteins. It includes EGFR, GAPDH, ESR1, CASP3, STAT3, BCL2, AKT1, CCND1, BCL2L1, PTGS2, HSP90AA1, PPARG, ERBB2, and RELA gene.

**Assessment of the central network.** The incorporation of 144 common targets into the STRING database facilitated the creation of a protein-protein interaction (PPI) network comprising 144 nodes and 1587 edges (Fig 2A). The data were exported as tab-separated values (TSV) and analyzed in Cytoscape 3.9.1 to evaluate the PPI network's topological properties. We identified 34 hub targets with degree values ≥31 (twice the median degree of 15.5; Fig 2B). The CytoNCA plugin (Cytoscape) analysis revealed median centrality metrics for 16 hub targets: degree centrality (28), betweenness centrality (4.32), closeness centrality (0.85), eigenvector centrality (0.17), local average connectivity (24.22), and network centrality (26.30). Ultimately, 16 core targets were identified, each exhibiting node scores that met or exceeded the

**Table 1. The top 14 components in Moxa smoke.**

| No. | Molecule name | CAS |
|---|---|---|
| 1 | Bis-(3,5,5-trimethylhexyl) phthalate | 14103-61-8 |
| 2 | 2-(dimethylamino) ethyl phenylacetate | 36882-00-5 |
| 3 | 3,7-Dimethyl-6,11-dodecadien-1-ol | 944384-91-2 |
| 4 | L-α-Terpineo | 10482-56-1 |
| 5 | Phthalic acid, bis(7-methyloctyl)ester | 20548-62-3 |
| 6 | 5-tert-Butylpyrogallol | 20481-17-8 |
| 7 | 4-Terpineol | 562-74-3 |
| 8 | α,β-Thujone | 76231-76-0 |
| 9 | 2',4'-Dimethoxyacetophenone | 829-20-9 |
| 10 | 1,4;3,6-Dianhydro-α-D-glucopyranose | 4451-31-4 |
| 11 | Camphor | 76-22-2 |
| 12 | Methenamine | 100-97-0 |
| 13 | Lupeol acetate | 1617-68-1 |
| 14 | 1-Tetracosanol | 506-51-4 |

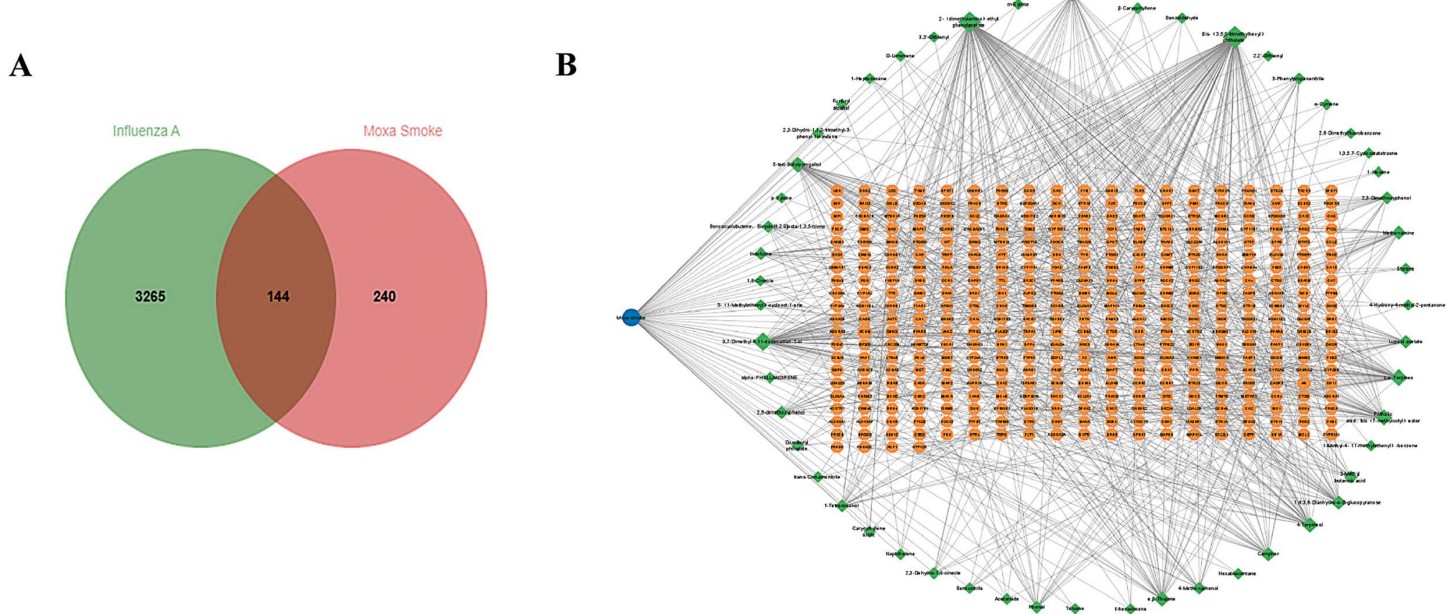

**Fig 1. Interactions between moxa smoke, ingredients, targets, and influenza A. (A)** Venn diagram of MS and IAV related targets. **(B)** PPI network of intersection targets. The blue circle represents the drug, the green diamond signifies the active ingredient, and the orange hexagon denotes the target. A larger area signifies a larger node, while a darker color represents a higher degree of association, and a lighter color indicates a lower degree of association.

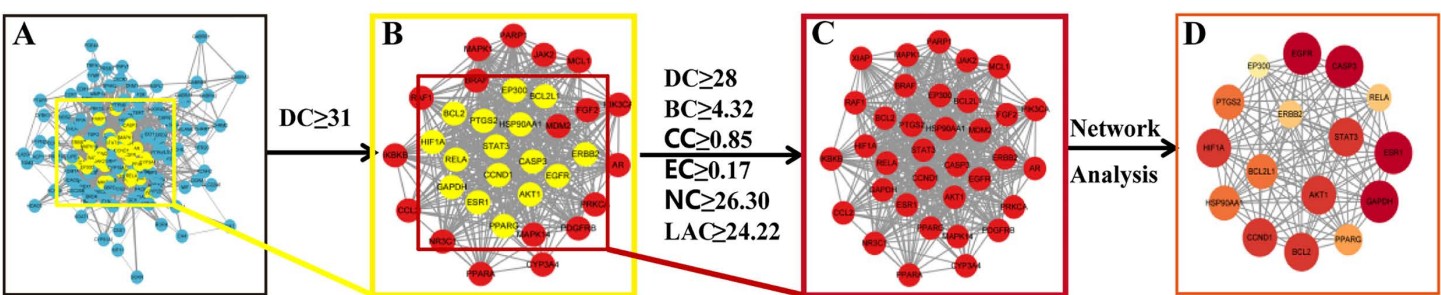

**Fig 2. Topological analysis of the shared target genes of Moxa smoke related to IAV. (A)** PPI network of the Moxa smoke-IAV crossover genes. **(B)** Network of 34 hub target genes based on degree values. **(C)** Central network of 16 core target genes based on DC, BC, CC, EC, LAC, and NC. **(D)** Visual analysis of the central network.

aforementioned median values (Table 2 and Fig 2C). Notable among these core targets are EGFR, GAPDH, ESR1, CASP3, STAT3, and BCL2, which are implicated as targets of moxa smoke in the treatment of H1N1.

Using the Network Analyzer plugin in Cytoscape, we conducted topological analysis of the core network, prioritizing nodes based on degree centrality values (Fig 2D). In this analysis, the degree was illustrated through variations in node color and size, with a higher degree being represented by nodes that were both darker in color and larger in size.

**GO and KEGG pathway enrichment analysis.** In order to provide a thorough and systematic examination of the potential mechanisms by which MS may exert therapeutic effects against H1N1 infection, we performed a GO enrichment analysis of its therapeutic targets at various levels, utilizing the Metascape database. This analysis encompassed BP,

**Table 2. The primary network data.**

| No. | Name of gene | DC | BC | CC | EC | LAC | NC |
|-----|-------------|-----|-----------|------------|------------|------------|------------|
| 1 | AKT1 | 89 | 2027.7372 | 0.71144277 | 0.2047481 | 23.797752 | 75.87847 |
| 2 | GAPDH | 89 | 2244.5273 | 0.7222222 | 0.20733726 | 24.494383 | 76.51658 |
| 3 | STAT3 | 74 | 1026.8423 | 0.6559633 | 0.19387819 | 25.864864 | 61.97606 |
| 4 | CASP3 | 72 | 708.2587 | 0.65296805 | 0.19321342 | 26.472221 | 59.172947 |
| 5 | BCL2 | 69 | 583.1304 | 0.6441441 | 0.19031101 | 26.753624 | 56.21779 |
| 6 | HSP90AA1 | 67 | 935.2286 | 0.63555557 | 0.17870875 | 24.298508 | 51.014618 |
| 7 | EGFR | 66 | 755.8988 | 0.63274336 | 0.18295231 | 26.181818 | 52.188255 |
| 8 | PPARG | 65 | 822.7298 | 0.63555557 | 0.16981791 | 23.138462 | 48.004017 |
| 9 | ESR1 | 61 | 540.7342 | 0.62173915 | 0.17330757 | 25.606558 | 46.701046 |
| 10 | HIF1A | 60 | 365.24796 | 0.6111111 | 0.1735665 | 25.966667 | 46.881924 |
| 11 | PTGS2 | 56 | 555.19165 | 0.6163793 | 0.15885974 | 23.321428 | 38.830227 |
| 12 | CCL2 | 52 | 736.81354 | 0.5884774 | 0.13034321 | 18.538462 | 36.072205 |
| 13 | EP300 | 49 | 265.01038 | 0.5697211 | 0.14960715 | 24.65306 | 37.5681 |
| 14 | ERBB2 | 49 | 169.61841 | 0.5836735 | 0.15547013 | 25.755102 | 36.806828 |
| 15 | CCND1 | 49 | 186.80348 | 0.5836735 | 0.15859148 | 27.265306 | 39.032272 |
| 16 | RELA | 47 | 492.0643 | 0.572 | 0.14662535 | 24.25532 | 33.72961 |

Degree centrality (DC), betweenness centrality (BC), closeness centrality (CC), eigenvector centrality (EC), network centrality (NC), and local average connectivity (LAC).

CC, and MF. We identified a total of 544 GO terms that reached statistical significance, which included 314 terms related to BP, 113 terms pertaining to CC, and 177 terms associated with MF. The top ten most enriched terms in BP, CC, and MF categories, ranked by gene count, were visualized using bar charts (Fig 3A). Our findings indicated that the targets of MS in the context of H1N1 treatment were primarily enriched in biological processes such as cellular responses to nitrogen compounds, regulation of monoatomic ion transport, modulation of hormone levels, regulation of the MAPK cascade, among other biological responses. Additionally, these targets were enriched in cellular components including the postsynapse, membrane rafts, receptor complexes, and the perinuclear region of the cytoplasm, among others. Furthermore, the molecular functions associated with MS targets included kinase binding, phosphotransferase activity, the ability to act as an acceptor for alcohol groups, heme binding, and various other molecular functions.

To uncover MS's anti-H1N1 action mechanisms, we identified relevant signaling pathways through KEGG analysis of its potential antiviral targets. This analysis revealed a total of 103 pathways that were statistically significant. The ten pathways with the highest number of associated genes are illustrated in a bubble graph (Fig 3B). Notably, the pathways related to Coronavirus disease-COVID-19, Epstein-Barr virus infection, Cytokine-cytokine receptor interaction, and Tuberculosis emerged as the four most prominent pathways.

## Molecular docking

The top 14 bioactive components were selected from the MS-ingredient-target-H1N1 network according to their degree centrality values. The compounds associated with MS is illustrated in Fig 4B. We performed molecular docking to evaluate binding energies of 14 selected bioactive compounds against 16 key target proteins (Fig 4B). To ensure a stable binding interaction between the compound and the protein, a screening criterion was established, necessitating a binding energy of less than −5.0 kcal/mol [29,30]. The results of this investigation indicated that the binding energies for the majority of interactions between the top 14 bioactive compounds and the 16 core target proteins were, in fact, below −5.0 kcal/mol. Furthermore, it was expected that a significant proportion of these 14 bioactive compounds exhibited a greater affinity

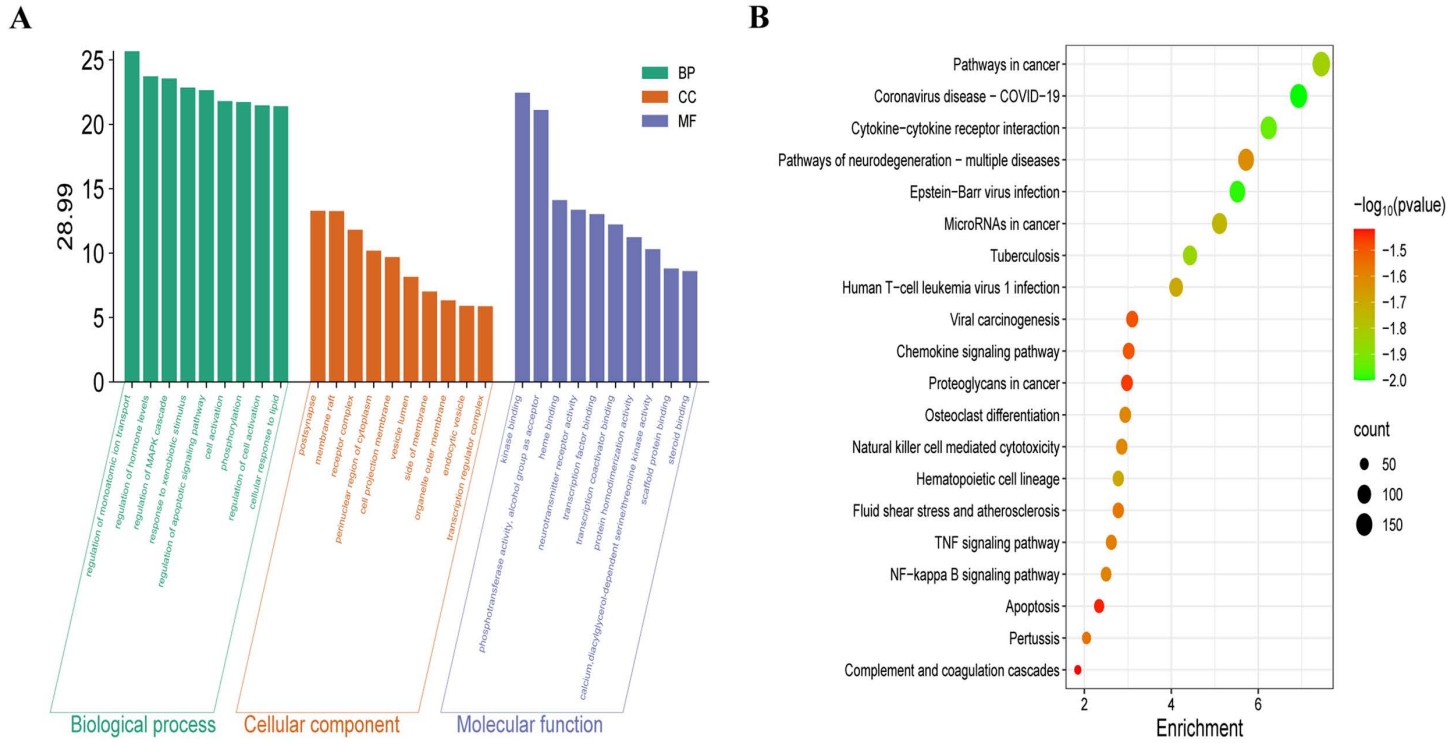

**Fig 3. Enrichment analysis. (A)** The GO enrichment analysis is presented, with fold enrichment represented on the y-axis and the corresponding terms on the x-axis, highlighting the ten principal results for BP, CC, and MF, respectively. **(B)** The KEGG pathway enrichment analysis, conducted using Metascape, is illustrated, with pathways displayed on the y-axis and false discovery rate (FDR) on the x-axis, while the color gradient indicates the P-values. The size of the bubbles corresponds to the number of genes that are enriched within each pathway.

for STAT3(PDB ID: 6njs), BCL2 (PDB ID: 8hts), PPARG (PDB ID: 9f7w), AKT1(PDB ID: 4gv1), CCND1(PDB ID: 2w96), BCL2L1(PDB ID: 7jgw), and PTGS2(PDB ID: 5f19), with binding energies less than −5.0 kcal/mol. Among the compounds, Bis-(3,5,5-trimethylhexyl) phthalate, lupeol acetate, L-α-Terpineo demonstrated more potent binding capacities than the others. The docking modes between Bis-(3,5,5-trimethylhexyl) phthalate, lupeol acetate, L-α-Terpineo and the afore mentioned proteins are represented in Fig 4A, hydrogen bond is represented by a dotted yellow line.

### MS-exposure does affect PR8-infected induced mice body weight loss

The body weight of each mouse was recorded on a daily basis, and the change in body weight was expressed as a percentage of their initial body weight, which was measured immediately before the commencement of the MS exposure on day 1 (Fig 5). Following the instranasal instillation of 12 PFU PR8, the rate of weight loss observed in the PR8+sham group mice was markedly greater compared to that of the MEM+sham group. Compared to MEM+sham mice, the average body weight of PR8+sham mice significantly decreased after ($p<0.01$, Fig 5). Compared to PR8+sham mice, the average body weight of PR8+LMS mice and PR8+HMS mice significantly decreased during the 4-day MS exposure protocol ($p<0.01$, Fig 5). The group that received PR8+HMS showed a more pronounced weight loss. But on the seventh day, the mice began to gain weight gradually.

### MS-exposure reduces the number of inflammatory cells in BALF from PR8-infected mice

The cellular inflammatory responses in the airways were evaluated by analyzing BALF collected from each mouse (Fig 6). PR8-infected mice exhibited a marked increase in the inflammatory cell count within their BALF when compared to those

 

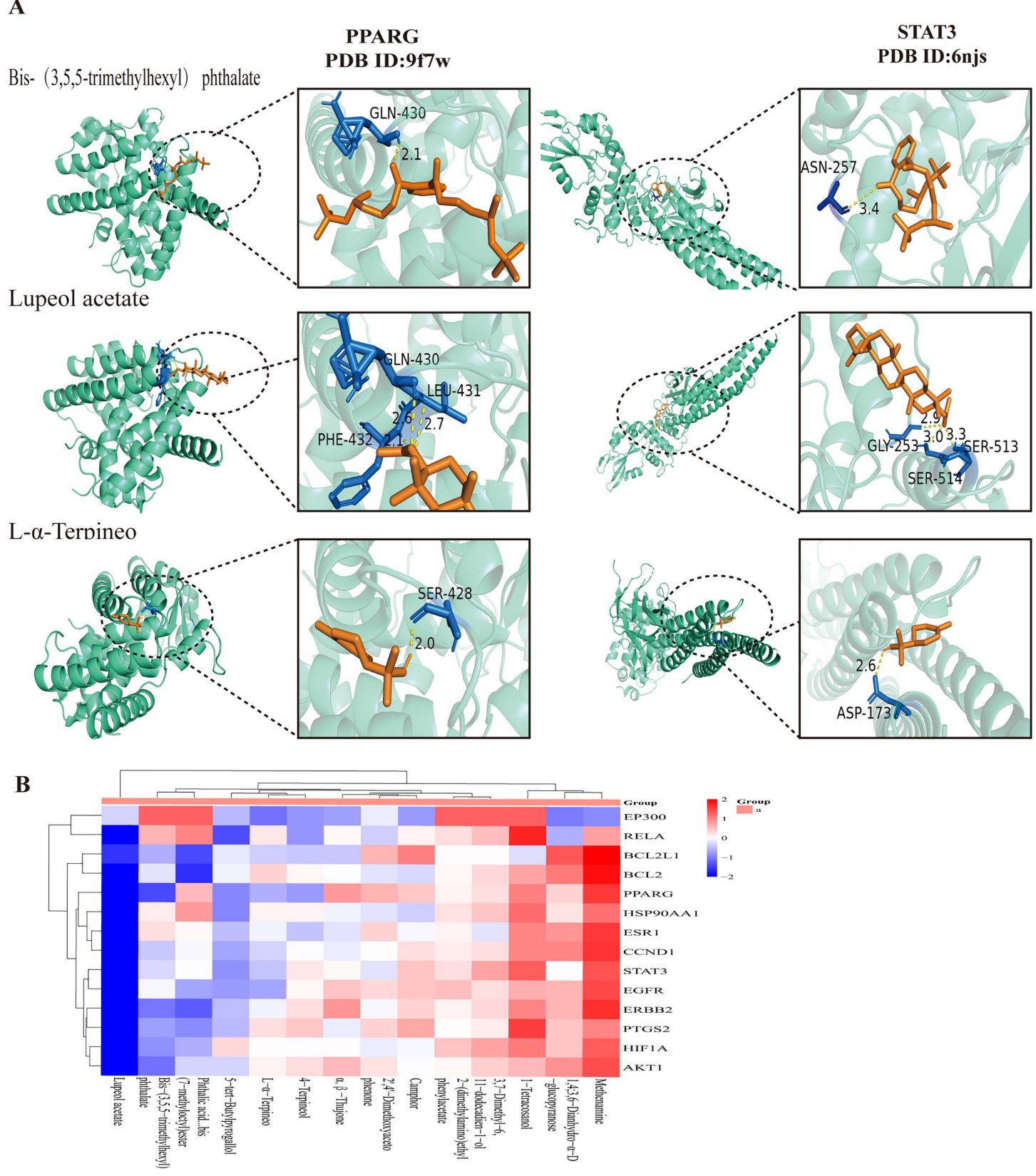

Fig 4. Molecular docking of bioactive components with key target proteins. (A) The docking configurations between the target proteins PPARG and STAT3 and the compounds Bis-(3,5,5-trimethylhexyl) phthalate, Lupeol acetate, and L-α-Terpineol, highlighting their respective lowest binding energies. (B) A heat map representing the molecular docking results of the top 14 bioactive components in relation to 16 core target proteins.

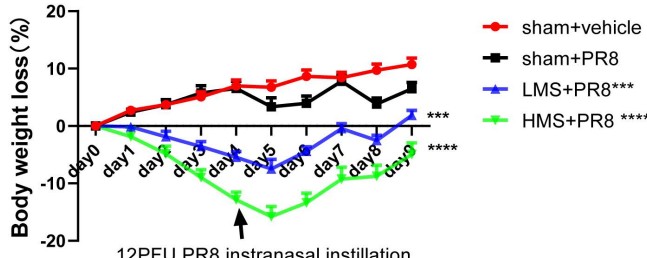

Fig 5. Effect of MS-exposure on body weight loss in PR8-infected mice. Mouse body weights were expressed as the percentage change in body weight and reported as the mean ± standard error of the mean (S.E.M.), with sample sizes varying from n = 7 to 9 mice per treatment group. A one-way ANOVA was performed to assess statistical significance, with thresholds set at **$p < 0.01$, ***$p < 0.001$, and ****$p < 0.0001$, compare to the PR8 + sham group.

in the MEM + sham group. Specifically, the total cell count, which encompasses macrophages, neutrophils, and lymphocytes, was significantly elevated in the BALF of the PR8 + sham mice relative to the MEM + sham mice ($p < 0.0001$). Notably, MS exposure resulted in a reduction in the total cell count, as well as in the counts of neutrophils and lymphocytes, in the BALF of PR8 + sham mice. However, MS exposure did not influence the macrophage count in the BALF of PR8 + sham mice. Among the various inflammatory cell types, the neutrophil count in PR8 + sham mice were significantly greater than that observed in normal mice. Conversely, a significant decrease in neutrophil count was observed, particularly during exposure to HMS. However, the inflammatory cell counts were decreased in mice exposed to MS compared to those in PR8 + sham mice.

## MS-exposure has the protective effects on lung injury during PR8-inlected

MS exposure attenuated inflammatory cell infiltration in the lung parenchyma. Histological examination of the lung tissue revealed that PR8 + sham mice exhibited extensive infiltration of inflammatory cells into alveolar, peribronchial, and perivascular lesions compared to normal mice. However, MS-exposed mice demonstrated a reduction in the alveolar histological alterations induced by PR8 instillation. Moreover, mice exposed to MS exhibited a notable decrease in the infiltration of inflammatory cells within the lung parenchyma when compared to the PR8 + sham group (Fig 7).

## MS-exposure reduces mRNA expression of the influenza virus nucleoprotein, cytokines, and chemokines in PR8-inflected mice

To confirm whether MS-exposure can decrease viral replication, we conducted a nucleoprotein (H1N1NP) mRNA expression test using TaqMan qPCR. Quantitative analysis revealed a marked upregulation of H1N1NP mRNA levels in PR8 + sham mice relative to MEM + sham controls($p < 0.01$). However, exposure to MS can reduce the mRNA expression of H1N1NP when compared to PR8 + sham mice (Table 3).

TaqMan qPCR analysis of whole lung tissue revealed MS-dependent changes in expression of inflammatory mediators (cytokines, chemokines, and Mmp12) in PR8-infected mice. Compared to MEM + sham mice, the PR8 + sham group showed marked upregulation of pro-inflammatory mediators, with significantly increased chemokine (Cxcl1, Cxcl2, Cxcl10, Ccl2; $p < 0.05$) and cytokine (Il-6, Il-1β, Tnf-α; $p < 0.01$) mRNA expression (Table 3). Conversely, the PR8-infected mice that

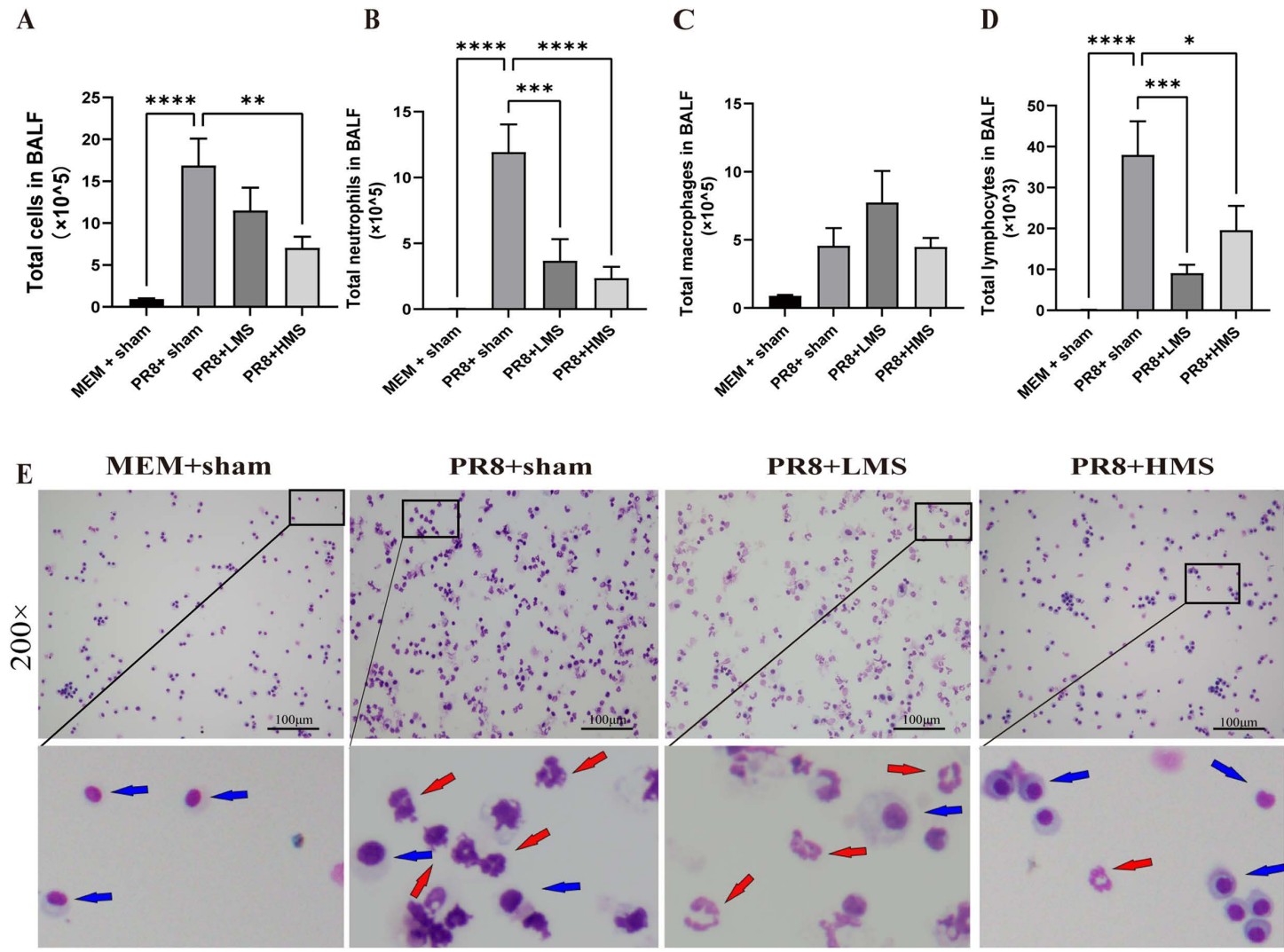

**Fig 6. Modulation of BALF Leukocyte Populations by MS in Influenza PR8-Infected Mice.** Cellularity is represented by the total population of cells**(A)**, neutrophils**(B)**, macrophages**(C)**, and lymphocytes**(D)**. **(E)** Representative images (original magnification × 200, scale bar = 100 μm) of Kwik-Diff-stained cytospin slides of BALF from each group. The blue arrow denotes macrophages, whereas the red arrow indicates neutrophils. Data are expressed as mean ± **S.**E.M for n = 7-9 per treatment group. One-way ANOVA was performed to assess statistical significance; *$p < 0.05$, **$p < 0.01$, ***$p < 0.001$, ****$p < 0.0001$, compare to the PR8 + sham group.

received MS exposure exhibited a notable reduction in the mRNA expression of cytokines (Il-6, Il-1β, and Tnf-α; $p < 0.01$) and chemokines (Cxcl1, Cxcl2, Cxcl10, and Ccl2; $p < 0.01$) when compared to the PR8 + sham group (Table 3). When compared to PR8 + sham mice, MS-exposure did not result in a reduction of Mmp12 expression. However, HMS-exposure increased the mRNA expression of Mmp12 ($p < 0.05$, Table 3).

## MS increased the expression of PPARγ and reduced phosphorylation of STAT3 in influenza infected mice

In order to validate the previously proposed targets and pathways associated with the anti-H1N1 effects of moxa smoke, we performed protein blotting to assess the expression levels of critical proteins, specifically phosphorylated STAT3 (p-STAT3), STAT3, and PPARγ. Infection with the PR8 strain led to a reduction in PPARγ protein levels within the lung

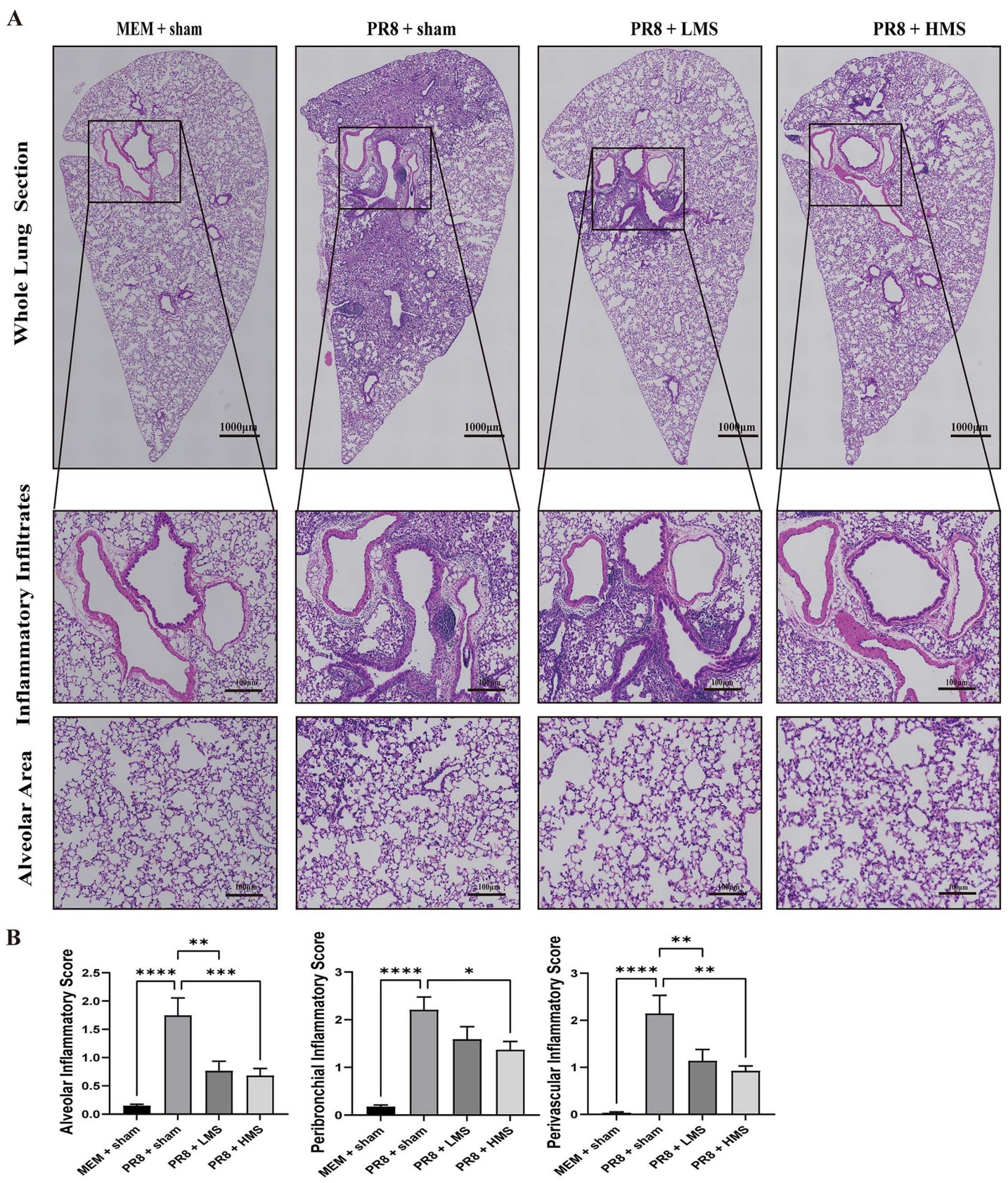

**Fig 7. Histopathological Evaluation of MS-Mediated Lung Protection Against Influenza PR8 Infection.** H&E-stained lung sections were collected from mouse that were euthanized on the fifth day following infection with the PR8 strain of influenza. **(A)** The upper four sections present representative composite images depicting the porta pulmonis region of the left lung lobes from four different groups of mice: MEM+sham, PR8+sham, PR8+LMS, and PR8+HMS, arranged from left to right. The lower four sections illustrate the presence of inflammatory infiltrates, while the final four sections depict the structural integrity and area of the pulmonary alveoli across the four groups of mice. **(B)** The assessment of inflammation scores in the alveolar, peribronchial, and perivascular regions. The data are expressed as mean±standard error of the mean (S.E.M) for a sample size of n=7-9 mice. Statistical significance was determined using one-way ANOVA, revealing significant differences among the groups (*$p<0.05$, **$p < 0.01$,***$p<0.001$,****$p<0.0001$).

**Table 3. Effect of MS exposure on lung proteins, cytokine, chemokine, and protease mRNA expression in MEM+sham, PR8+sham, PR8+LMS, and PR8+HMS mice.**

| Gene | MEM+sham | PR8+sham | PR8+LMS | PR8+HMS |
|---|---|---|---|---|
| Nucleoprotein | | | | |
| H1N1NP | 1.250±0.3885 | 5106±1948▲▲ | 764.2±457.1** | 1057±544.4* |
| Cytokines | | | | |
| Il-6 | 1.160±0.2391 | 23.69±8.720▲▲ | 3.719±1.494** | 6.541±3.068* |
| Tnf-α | 1.115±0.1701 | 4.781±1.578▲▲ | 1.694±0.4858* | 1.437±0.2935* |
| Il-1β | 1.163±0.2440 | 2.473±0.5605▲ | 1.121±0.1941* | 1.028±0.1718* |
| Gm-csf | 1.107±0.1671 | 1.128±0.2032 | 0.9367±0.1314 | 0.9619±0.09051 |
| Chemokines | | | | |
| Cxcl1 | 1.132±0.1960 | 5.924±2.158▲▲ | 1.863±0.4177* | 2.117±0.3964* |
| Cxcl2 | 1.090±0.1649 | 5.344±2.038▲ | 1.318±0.4159* | 1.589±0.6526* |
| Cxcl10 | 1.039±0.09504 | 77.88±23.56▲▲▲ | 18.30±11.10* | 18.10±11.15* |
| Ccl2 | 1.054±0.1173 | 28.72±8.672▲▲ | 8.002±3.944* | 10.25±5.295* |
| Proteases | | | | |
| Mmp12 | 1.183±0.2344 | 0.6739±0.1215 | 1.565±0.1186 | 2.215±0.4631* |

Gene expression was assessed concurrently utilizing TaqMan quantitative PCR under uniform conditions. The results are expressed as fold changes in comparison to MEM+sham mice, with normalization to the expression of the housekeeping gene, Gapdh. The data are reported as the mean plus standard deviation for triplicate reactions derived from 7–9 individual mouse lung samples. One-way ANOVA was performed to determine statistical significance. ▲$p<0.05$, ▲▲$p<0.01$,▲▲▲$p<0.001$ versus the MEM+sham group; *$p<0.05$, **$p<0.01$, ** $p<0.001$ versus the PR8+sham group.

tissue of mice; however, this alteration did not reach statistical significance. Conversely, PPARγ levels increased in mice following MS treatment. Additionally, moxa smoke treatment reduced the p-STAT3 protein level, while STAT3 protein levels remained unchanged (Fig 8).

## MS alleviates the recruitment of pro-inflammatory cells by activation of PPARγ in H1N1-infected mice

To determine whether MS alleviates airway inflammation and reduces viral load via PPARγ activation, we compared the effects of HMS with the PPARγ agonist pioglitazone, as well as HMS combined with the PPARγ inhibitor GW9662, in H1N1-infected mice.

The results showed that, compared with the model mice, HMS treatment decreased the population of BALF cells, especially neutrophils and lymphocytes (Fig 9A), and reduced the mRNA expression of NP, IL-6, TNF-α, and IL-1β (Fig 9B) in the lung tissue of H1N1-infected mice, consistent with previous findings. When the mice were given GW9662 prior to HMS treatment, the cell population in the BALF, NP, and cytokines were reduced compared to the model mice; however, the number of BALF cells was higher than in mice treated with HMS alone, without GW9662 (Fig 9A and 9B). Pioglitazone treatment alone reduced the population of BALF cells but did not affect NP or cytokine levels in model mice. Regarding STAT3 phosphorylation, neither pioglitazone nor GW9662 combined with HMS reduced the level of p-STAT3 (Fig 9C).

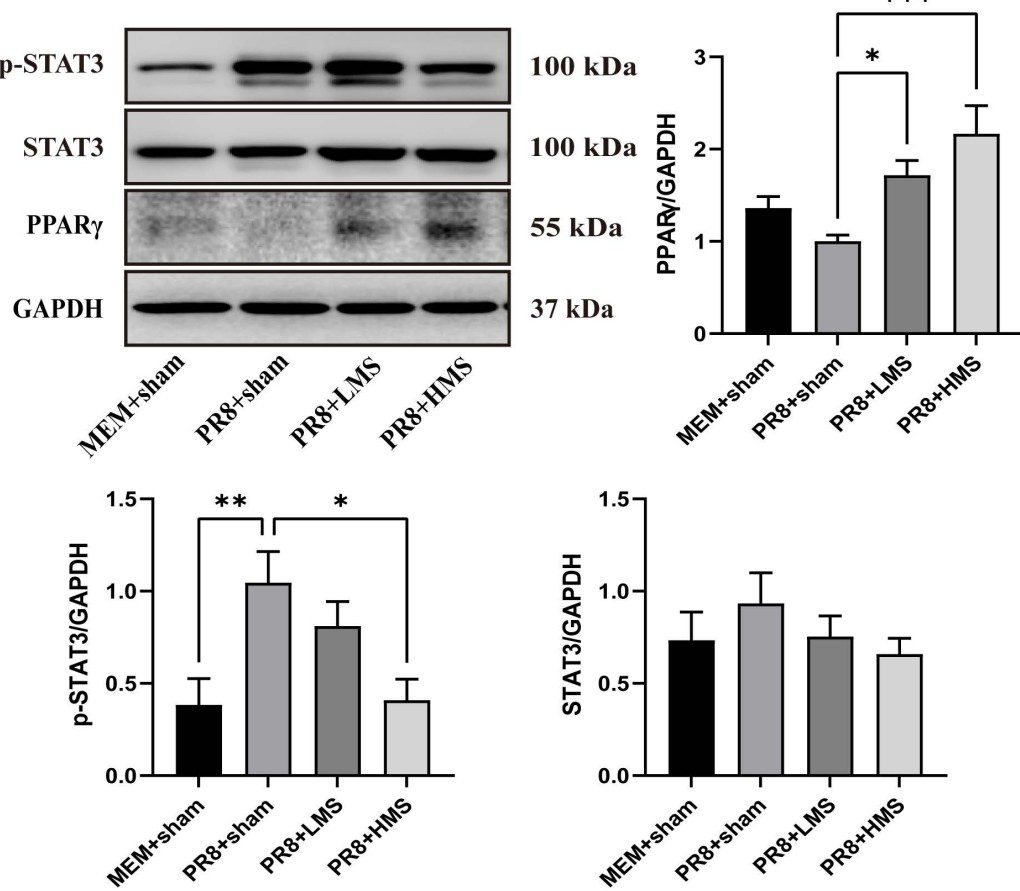

**Fig 8. MS increased the expression of PPARγ and reduced phosphorylation of STAT3.** The measurement of p-STAT3, STAT3, and PPARγ through western blot analysis (n = 6). All data are expressed as means ± standard error of the mean (SEM). Statistical significance was determined in comparison to the model group, with thresholds set at *$p < 0.05$, **$p < 0.01$, and ***$p < 0.001$.

## Safety assessment of MS

MS exposure for 4 days in H1N1-infected mice: Table 4 presents the serum biochemical parameters of H1N1 infected mice subjected to moxa smoke exposure for a duration of 4 days. Analysis of the blood biochemical markers revealed that, compared to the PR8 group, both the LMS and HMS group exhibited reductions in ALT and AST levels. The LMS also reduced Urea of the H1N1 infected mice.

MS exposure for 4 weeks in normal mice: Table 5 presents the blood biochemical parameters of mice subjected to moxa smoke exposure for a duration of 4 weeks. Analysis of the blood biochemical markers revealed that, relative to the sham group, the LMS group exhibited reductions in AST and Urea levels in male mice. And increased the TCO2 in male mice. In contrast, the HMS group demonstrated decreased Urea concentrations, alongside elevated TCO2 levels in both male and female mice, when compared to the sham group. HMS also reduced the level of ALT in male mice compared to sham mice.

Fig 10A showed the organ weight ratio of sham mice after MS exposure. Mice exposed to LMS and HMS both showed increase in the ratios of lung/body weight. Female mice exposed to HMS exhibited an increased kidney-to-body weight ratio.

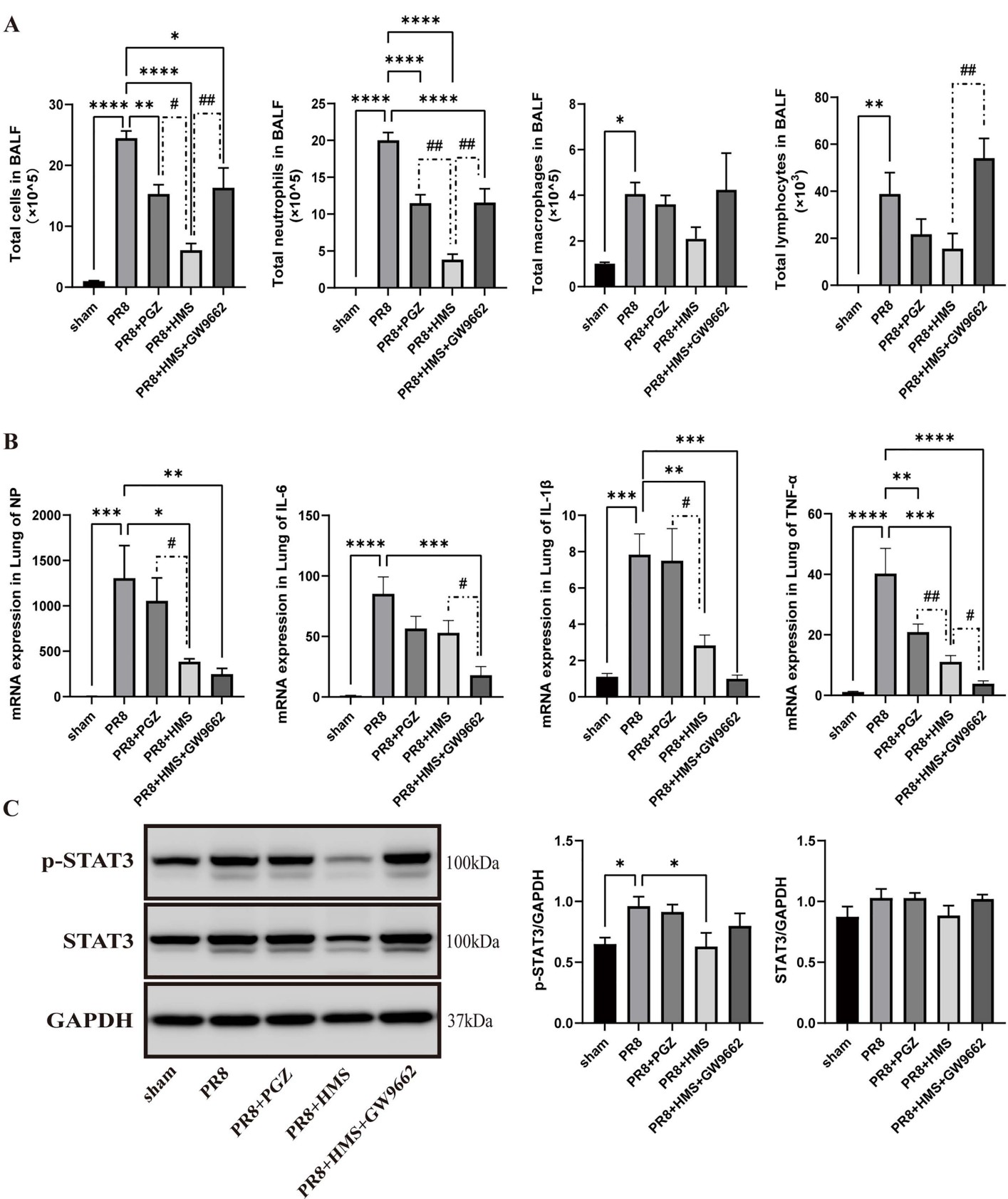

**Fig 9. MS alleviates the recruitment of pro-inflammatory cells by activation of PPARG in H1N1-infected mice. (A)** Total cell number and differential cell count in BALF. **(B)** mRNA expression of NP and cytokines. **(C)** p-STAT3 and STAT3 protein level. Data are expressed as mean±**S**.E.M for n=6 per treatment group. One-way ANOVA was performed to assess statistical significance: $*p<0.05$, $**p<0.01$, $***p<0.001$, $****p<0.0001$, compare to PR8 group; $\#p<0.05$, $\#\#p<0.01$, compare to PR8 group.

**Table 4. Serum biochemistry parameters of H1N1-infected mice treated with inhaled MS for 4 days.**

| Serum biochemistry | PR8 | PR8+LMS | PR8+HMS |
|---|---|---|---|
| ALT/U·L$^{-1}$ | 39.14±2.0 | 26.38±0.68**** | 27.88±0.74**** |
| AST/U·L$^{-1}$ | 116.4±9.26 | 83.75±3.99** | 72.75±4.2*** |
| AST/ALT | 3.27±0.30 | 3.2±0.19 | 2.613±0.18 |
| Urea/mol·L$^{-1}$ | 6.46±0.39 | 4.838±0.24** | 6.05±0.43 |
| Cr/μmol·L$^{-1}$ | 5.86±0.26 | 5.5±0.5 | 5±0.27 |
| TCO2/mmol·L$^{-1}$ | 20.71±0.74 | 20.53±0.47 | 20.98±0.31 |

Data are expressed as mean±S.E.M for n=7–8 male mice per treatment group. One-way ANOVA was performed to assess statistical significance; $**p<0.01$, $***p<0.001$, $****p<0.0001$.

**Table 5. Serum biochemistry parameters of mice treated with inhaled MS for 4 weeks.**

| Serum biochemistry | Gender | sham | LMS | HMS |
|---|---|---|---|---|
| ALT/U·L$^{-1}$ | Female | 29±0.58 | 26.33±1.2 | 25.75±1.11 |
| | Male | 32.8±3.09 | 28.8±1.02 | 25.25±1.03* |
| AST/U·L$^{-1}$ | Female | 78.25±4.25 | 82.67±6.69 | 73.5±1.76 |
| | Male | 78.4±1.17 | 65.4±3.75* | 70.75±6.63 |
| AST/ALT | Female | 2.7±0.11 | 3.13±0.15 | 2.85±0.17 |
| | Male | 2.44±0.19 | 2.26±0.12 | 2.78±0.18 |
| Urea/mol·L$^{-1}$ | Female | 6.43±0.38 | 5.33±0.23 | 4.95±0.26** |
| | Male | 8.4±0.39 | 5.64±0.22**** | 5.38±0.15**** |
| Cr/μmol·L$^{-1}$ | Female | 6.48±0.67 | 6.67±0.33 | 7.75±0.25 |
| | Male | 5.98±0.56 | 5.18±0.21 | 4.95±0.03 |
| TCO2/mmol·L$^{-1}$ | Female | 17.3±0.69 | 18.93±0.07 | 18.85±0.35* |
| | Male | 18.2±0.43 | 19.88±0.35* | 21.03±0.44*** |

n=3–4 for female, n=4–5 for male. Data are expressed as mean±S.E.M. One-way ANOVA was performed to assess statistical significance; $*p<0.05$, $**p<0.01$, $***p<0.001$, $****p<0.0001$.

Histopathological examination (Fig 10B) showed that (1) Compared to control mice, more macrophages were observed in lungs exposed with both LMS and HMS, and the alveolar septum structure was normal. (2) Compared to control mice, renal tubular epithelial cells were completely, there were no obviously granules and vacuolar degeneration. However, in female mice which exposed to HMS, there were slightly interstitial edema. (3) The liver, kidney, testis, and ovary tissues after four weeks of MS exposure revealed no discernible pathological abnormalities compared to the sham controls.

## MS promotes MMP12 expression, but has not yet led to the development of emphysema or abnormal collagen deposition in 4 weeks

The MMP12 mRNA in lung tissue were increased after both 4-day MS exposure and 4-week MS exposure compared to sham mice. Moreover, compared to the mice exposed to MS for 4 days, the expression level of MMP12 in the lung tissue

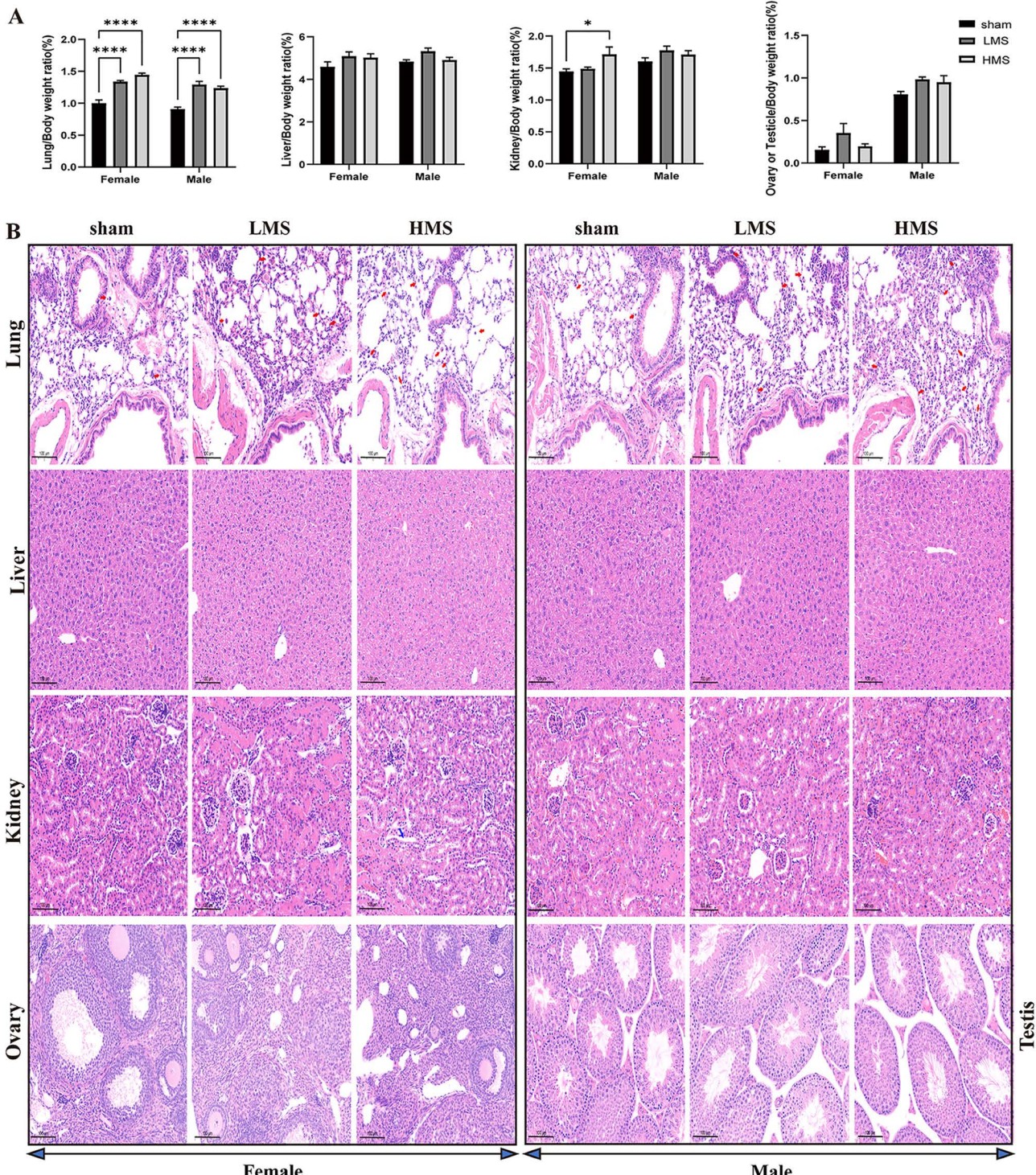

**Fig 10. The Toxicological Effects of MS on Mice. (A)** The origan weight/body weight ratio (%). **(B)** Histopathological images of organ tissues. The red arrow denotes the macrophages within the pulmonary tissue, while the blue arrow identifies the interstitial space. Data are expressed as mean ± S.E.M for n = 6 per treatment group. One-way ANOVA was performed to assess statistical significance; *$p < 0.05$, ****$p < 0.0001$.

of mice exposed to MS for 4 weeks is higher. This result indicates that the increase in lung MMP12 levels induced by MS is time-dependent (Fig 11A).

To clarify whether short-term exposure to MS causes abnormal lung collagen levels or emphysema formation in influenza virus-infected mice and normal mice with long-term exposure to MS, we measured lung collagen level by hydroxyproline measurement and Masson's trichrome staining, and the emphysema formation (MLI index) was also detected in influenza-infected mice exposed to MS for 4 days and normal mice exposed to MS for 4 weeks. The results demonstrated that the collagen level and MLI index didn't change after 4-day and 4-week exposure of MS (Fig 11 B–11D).

## Discussion

Influenza is a highly transmissible viral infection predominantly attributed to Influenza A viruses, which impacts the respiratory system and has the potential to result in significant morbidity. The emergence of viral variability and the development of drug resistance underscore the pressing necessity for the discovery and development of novel antiviral therapeutics [31]. Moxa smoke (MS) exhibits broad antiviral activity and immune-modulating effects, enhancing immune function while suppressing inflammation [13,32–34]. Through network pharmacology, we identified the core targets of MS against H1N1-infected by constructing disease-gene-target-drug interaction networks to elucidate its multi-target mechanisms [34]. This research combines network pharmacology with animal experimentation to comprehensively examine the anti-H1N1 infection properties of MS and to confirm its therapeutic targets. Moxa smoke demonstrated a reduction in the infiltration of inflammatory cells within the lungs, along with a decrease in the expression levels of pro-inflammatory cytokines and chemokines in mice infected with the PR8 strain. Additionally, it decreased viral loads in vivo.

In the current study, 52 components were identified in moxa smoke. There are many essential oil components in moxa smoke, including camphor [35–38], 4-Terpineol [37,39,40], L-alpha-terpineol [39], α,β-thujone [41–42], D-limonene [41,43,44], caryophyllene oxide [45–46], β-caryophyllene [45–48], 2,3-dehydro-1,8-cineole [49], and 1,8-cineole [50–55], which have been shown to possess antiviral effects. And Lupeol acetate demonstrated an anti-inflammatory effect [56]. In the GO enrichment analysis, the terms "inflammatory response" and "response to xenobiotic stimulus" exhibited significant enrichment. Furthermore, the KEGG analysis revealed that multiple signaling pathways associated with the treatment of H1N1 influenza, including the NF-kappa B signaling pathway, the TNF signaling pathway, and additional pathways related to apoptosis, were implicated in moxa smoke. The core targets of PPI analysis include STAT3, BCL2, RELA, CASP3, AKT1, CCND1, BCL2L1, PTGS2, and PPARG. Moxa smoke has the potential to induce apoptosis in A549 alveolar type II epithelial cells [57].

Infectious agents, including influenza A virus (IAV), directly affect and impair lung epithelial cells and alveolar macrophages, which subsequently triggers immune responses and leads to acute lung injury [58]. During the early stages of infection, monocytes/macrophages and neutrophils represent the primary innate immune cells that are mobilized to the alveolar area [58]. Neutrophils, as a crucial element of the immune response following a viral infection, possess the ability to swiftly relocate from the bloodstream to the infected pulmonary tissue. Within this environment, they engage in a variety of antimicrobial activities, which encompass phagocytosis, the generation of reactive oxygen species (ROS), degranulation, and the formation of neutrophil extracellular traps (NETs) [59]. While appropriate neutrophil activity facilitates infection clearance, excessive activation can exacerbate tissue damage in severe cases [60]. Given that pulmonary neutrophil infiltration correlates with disease severity in viral infections [61], controlling neutrophil-mediated inflammation is a crucial therapeutic strategy for H1N1 infection. Our findings demonstrate that moxa smoke effectively reduces neutrophil accumulation in BALF.

While neutrophil activation serves as a critical defense mechanism against infections, it also contributes to tissue damage and inflammatory pathology [62]. This process is mediated by chemotactic signals, particularly the marked elevation of CXCL1 and CCL2, which facilitate the rapid migration of neutrophils to inflamed lung tissue. Our investigation revealed that H1N1 infection triggers a substantial upregulation of pro-inflammatory mediators in pulmonary tissue, including Il-6,

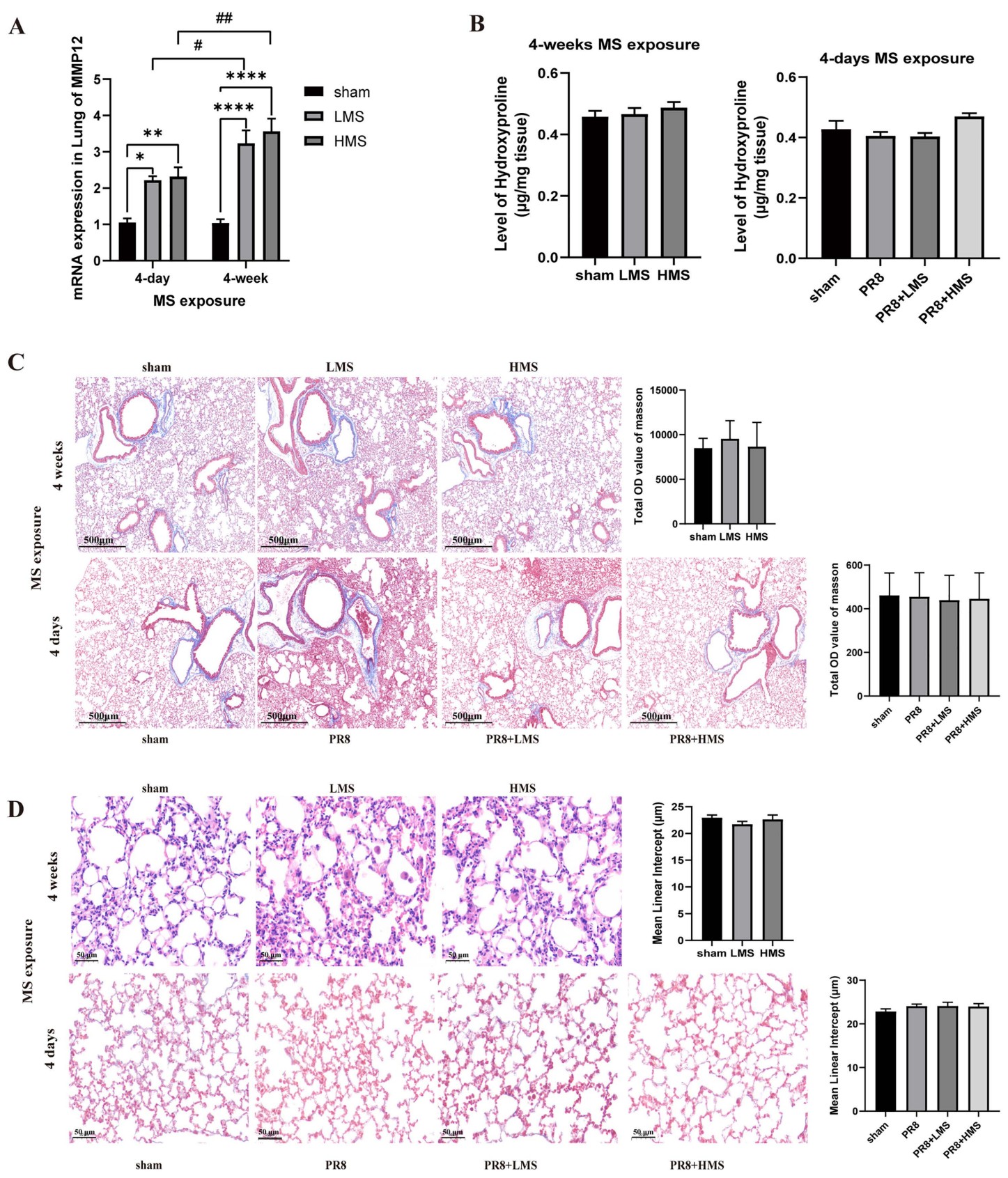

**Fig 11. The effects of MS on MMP12 mRNA Expression, Pulmonary Collagen and Lung Tissue Injury. (A)** The mRNA expression of MMPI2 in lung tissue of normal mice after 4 days and 4 weeks MS exposure (n=7-10). **(B)** Hydroxyproline level in lung tissue (n=7-10). **(C)** Masson's trichrome staining (n=6). **(D)** MLI index (n=6). Data are expressed as mean±**S.**E.M. One-way ANOVA was performed to assess statistical significance; *$p<0.05$, **$p<0.01$, ****$p<0.0001$.

Il-1β, Tnf-α, Cxcl1, Cxcl 2, Cxcl 10, and Ccl2--all implicated in the pathogenesis of acute lung injury. Importantly, moxa smoke treatment demonstrated significant therapeutic potential by effectively reducing the levels of these cytokines and chemokines, thereby attenuating pulmonary inflammation.

STAT3, a multifunctional transcription factor, is integral to numerous biological processes, encompassing inflammation, immune regulation, angiogenesis, and the maintenance of cellular homeostasis [63]. Previous research by Liu et al. demonstrated that IAV infection triggers rapid phosphorylation of STAT3 at tyrosine 705 (Y705) [63]. Importantly, the suppression of STAT3 phosphorylation in H1N1 PR8-infected bone marrow-derived macrophages (BMDMs) has been shown to significantly decrease viral replication [64]. Consistent with these findings, our study demonstrates that moxa smoke treatment effectively reduces both viral load and levels of phosphorylated STAT3 protein in lung tissue.

Peroxisome proliferator-activated receptor gamma (PPARγ) operates as a nuclear transcription factor that primarily forms heterodimers with the retinoid X receptor (RXR). This heterodimeric complex facilitates the recruitment of transcriptional co-regulators to PPAR response elements, thereby modulating the expression of genes associated with adipogenesis, lipid metabolism, and inflammatory processes [65]. PPARγ serves as a crucial regulator of both M2 macrophage polarization and inflammatory mediator overproduction [63–67]. Experimental evidence demonstrates that pharmacological activation of PPARγ, whether administered prophylactically or therapeutically, significantly improves survival outcomes in mice infected with IAV [68–71]. Under physiological conditions, pulmonary PPARγ expression is primarily localized in macrophages, where it supports their proper differentiation and facilitates recovery following influenza infection [72–74]. However, influenza infection triggers a type I interferon receptor-dependent downregulation of PPARγ in alveolar macrophages [75–76]. Notably, our current findings reveal that moxa smoke treatment similarly increased PPARγ protein levels in lung tissue.

Does moxa smoke exert its antiviral and anti-inflammatory effects via PPARγ? Our experimental findings indicate that when PPARγ is inhibited by GW9662, the inhibitory effect of moxa smoke on the recruitment of immune cells to the lungs of H1N1-infected mice is markedly diminished, particularly with regard to neutrophils and lymphocytes. This demonstrates that moxa smoke reduces the infiltration of inflammatory cells in the lungs of H1N1-infected mice, and that this effect is dependent on PPARγ.

Interestingly, however, PPARγ agonists failed to reduce the viral load in mouse lungs, whereas moxa smoke significantly decreased the viral load in lung tissue. Following inhibition by PPARγ antagonists, the suppression of viral replication by moxa smoke appeared more pronounced than with moxa smoke alone. This indicates that the reduction in pulmonary viral load caused by moxa smoke does not depend on PPARγ. Conversely, inhibiting PPARγ enhances the viral clearance effect of moxa smoke. The reason may be that the suppression of immune cell recruitment by PPARγ activation partially impedes influenza virus clearance. As previously discussed, certain components within moxa smoke exhibit antiviral activity independent of PPARγ. This explains why moxa smoke exerts a stronger inhibitory effect on immune cells than PPARγ agonists, and why PPARγ's role in reducing pulmonary immune cells persists even when PPARγ is inhibit.

Previous literatures indicates that activation of PPARγ can inhibit STAT3 phosphorylation [77–78]. In this study, PPARγ agonists did not markedly reduce p-STAT3 levels. This is because influenza virus infection elevates p-STAT3 levels, thereby masking the inhibitory effect of PPARγ agonists on p-STAT3. When the viral load decreases, inhibiting PPARγ increases p-STAT3 levels; hence, p-STAT3 levels in the moxa smoke+GW9662 group were higher than those in the moxa smoke group. However, according to our results, the primary inflammatory cytokines—IL-6, TNF-α, and IL-1β—appear to correspond with the level of viral load and are not primarily regulated by p-STAT3.

Although studies indicate that exposure to MS does not significantly affect the health of physicians [79–80], the presence of harmful substances in moxa smoke [81] and evidence from animal studies suggesting that it impairs respiratory function [82] make it imperative to investigate its safety profile. This study first examined biochemical parameters in influenza-infected mice following short-term exposure to moxa smoke. The results indicated no apparent abnormalities in liver or kidney function. Although MMP12 expression increased in the lungs, no emphysema or abnormal collagen deposition was observed, nor was carbon dioxide retention detected in the blood. In contrast, normal mice exposed to moxa smoke for four weeks exhibited increased lung tissue weight, heightened inflammatory cell infiltration in the lungs, elevated pulmonary MMP12 expression (without inducing pulmonary emphysema or changes in collagen deposition) and mild oedema of renal tissue in the high-dose moxa smoke exposure group. These findings indicate that short-term exposure to moxa smoke is safe, but prolonged exposure causes significant damage to the lungs and renal function. Furthermore, compounds in moxa smoke, such as phenol, benzonitrile, and 1,1,2,2-tetrachloroethane, can cause respiratory system damage, alongside hepatotoxicity, nephrotoxicity, metabolic disruption, and carcinogenic risks [83–85]. Based on the findings of this study, short-term exposure to moxa smoke for the treatment of influenza virus is recommended as both effective and safe. However, prolonged exposure should be avoided wherever possible, with particular restrictions for the elderly, children, and individuals with underlying medical conditions. Furthermore, animal studies often burn moxa sticks in relatively confined spaces to control smoke concentration, which paradoxically exacerbates incomplete combustion. The resulting increase in harmful substances heightens toxicity, creating a discrepancy with real-world clinical settings. This underscores that, in clinical moxa smoke therapy, alongside controlling exposure duration, attention must be paid to environmental ventilation to minimise incomplete combustion. It is also recommended that occupational exposure standards for moxa smoke be established.

## Conclusion

Moxa smoke increased the expression of PPARγ in the lungs of mice infected with influenza A virus and suppressed influenza A virus-induced inflammatory cell infiltration, which depends on the activation of PPARγ. Simultaneously, it reduced viral load through PPARγ-independent mechanisms. Short-term exposure to moxa smoke did not cause significant impairment of pulmonary, hepatic, or renal function; however, prolonged exposure may result in respiratory and renal dysfunction, potentially leading to more severe adverse effects.

## Supporting information

**S1 File. Raw images.**
(PDF)

**S2 File. Raw Dataset.**
(XLSX)

## Author contributions

**Conceptualization:** Long Fan, Xuhua Yu.

**Data curation:** Ting Cao.

**Formal analysis:** Ting Cao, Xuhua Yu.

**Funding acquisition:** Xuhua Yu.

**Investigation:** Ting Cao, Wenchao Pan, Ziyao Liang, Jingyu Quan, Miaona Zhang, Huameng Li.

**Methodology:** Ting Cao, Ziyao Liang, Jingyu Quan, Xuhua Yu.

**Validation:** Ting Cao, Xuhua Yu.

**Visualization:** Ting Cao, Wenchao Pan.

**Writing – original draft:** Ting Cao.

**Writing – review & editing:** Xuhua Yu.

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
