## [Decision Letter · Decision Letter 0]

4 Jul 2025

Investigation of the Effect and Mechanisms of Moxa Smoke in the Treatment of Influenza A Virus (IAV) Infection

PLOS ONE

Dear Dr. Yu,

Thank you for submitting your manuscript to PLOS ONE. After careful consideration, we feel that it has merit but does not fully meet PLOS ONE’s publication criteria as it currently stands. Therefore, we invite you to submit a revised version of the manuscript that addresses the points raised during the review process.

We look forward to receiving your revised manuscript.

Kind regards,

Md Bashir Uddin, PhD

Academic Editor

PLOS ONE

Journal Requirements:

“Guangzhou Key Laboratory of Traditional Chinese Medicine for the Prevention and

Treatment of Chronic Cough and Dyspnea (Grant Number 2023A03J0226),

the double world-class and high-level university discipline collaborative innovation

team project of Guangzhou University of Chinese Medicine (Grant Number

2021XK27).”

4. We note that your Data Availability Statement is currently as follows: All relevant data are within the manuscript and in Supporting Information files.

5. Please include a copy of Table 4 which you refer to in your text on page 22.

Reviewers' comments:

Reviewer's Responses to Questions

**Comments to the Author**

1. Is the manuscript technically sound, and do the data support the conclusions?

Reviewer #1: Yes

Reviewer #2: No

2. Has the statistical analysis been performed appropriately and rigorously?

Reviewer #1: Yes

Reviewer #2: No

3. Have the authors made all data underlying the findings in their manuscript fully available?

Reviewer #1: No

Reviewer #2: Yes

4. Is the manuscript presented in an intelligible fashion and written in standard English?

Reviewer #1: Yes

Reviewer #2: No

Reviewer #1: This study explores the antiviral and anti-inflammatory effects of moxa smoke (MS) against H1N1 infection using network pharmacology, molecular docking, and in vivo experiments. The integration of traditional Chinese medicine (TCM) with modern pharmacological approaches is commendable, offering novel insights into MS’s potential therapeutic applications. However, while the research demonstrates scientific rigor in certain aspects, it contains critical gaps, particularly regarding clinical safety implications and mechanistic clarity, which must be addressed for publication.

Strengths and Innovations

1.Multidisciplinary Approach to Mechanism DiscoveryThe study employs network pharmacology to systematically identify MS components (52 compounds) and their potential targets (384 proteins), uncovering 16 core targets (e.g., STAT3, PPARγ) via PPI network analysis. This “multi-component, multi-target” framework aligns with TCM’s holistic philosophy and provides a robust basis for understanding MS’s complex actions. The molecular docking validation of stable binding between MS compounds (e.g., Bis-(3,5,5-trimethylhexyl) phthalate) and key targets further strengthens the mechanistic hypothesis.

2. In Vivo Validation of Antiviral and Anti-Inflammatory EffectsUsing a PR8-infected mouse model, the study demonstrates that MS reduces viral load (H1N1NP mRNA), alleviates pulmonary inflammation (e.g., decreased BALF leukocytes, IL-6/TNF-α expression), and modulates PPARγ/STAT3 pathways. These findings provide experimental evidence for MS’s therapeutic potential, particularly its ability to suppress cytokine storms—a critical pathology in severe influenza.

3. Translational Relevance to TCM PracticeBy focusing on MS, a byproduct of moxibustion with historical use in infection prevention, the study bridges traditional practices with modern science. The identification of PPARγ as a key regulator offers a mechanistic explanation for MS’s anti-inflammatory effects, potentially guiding the development of targeted TCM interventions.

Critical Weaknesses and Safety Concerns

1. Inadequate Assessment of Clinical Safety Risks

Occupational Exposure Risks for Acupuncturists:The manuscript fails to address the chronic toxicity of MS, a major concern for acupuncturist) and patients. Existing evidence (cited in prior reviews) shows that long-term exposure to moxa smoke is associated with increased incidence of chronic pharyngitis, allergic reactions, and potential pulmonary fibrosis. For example, elevated MMP12 expression in MS-treated mice (Table 3) indicates ECM degradation, a hallmark of emphysema and pulmonary fibrosis. The study’s failure to discuss this finding’s clinical relevance constitutes a significant oversight.

Systemic Toxicity of Smoke Components:MS contains known toxicants, including PM2.5/PM10, aldehydes (e.g., formaldehyde), and polycyclic aromatic hydrocarbons (PAHs). While the study excludes “toxic gaseous compounds,” it provides no data on the safety of identified components (e.g., phthalates, terpenoids) in long-term use. For instance, phthalates are endocrine disruptors linked to metabolic and reproductive disorders, and PAHs are carcinogenic. The absence of toxicity profiling (e.g., hepatic/renal function assays, genotoxicity tests) undermines the safety narrative.

2. Mechanistic Gaps and Incomplete Data Interpretation

Contradictory Findings with MMP12 Upregulation:The study reports that high-dose MS (HMS) significantly upregulates MMP12 mRNA in lung tissue (p<0.05), yet dismisses this as a “potential risk” without mechanistic exploration. MMP12’s role in alveolar wall destruction and fibrosis is well-established; its induction by MS contradicts the claimed “protective effect” on lung injury. The authors must clarify whether this is a dose-dependent effect or an experimental artifact, ideally through immunohistochemistry or functional assays (e.g., ECM degradation markers).

Unclear Dynamics of PPARγ/STAT3 Pathways:While MS upregulates PPARγ and reduces p-STAT3, the study does not address how these changes balance antiviral immunity and inflammatory suppression. For example, PPARγ overactivation may impair macrophage antiviral function, while STAT3 inhibition could compromise viral clearance. Long-term outcomes (e.g., viral persistence, secondary infections) are not evaluated, limiting conclusions about therapeutic windows.

3. Experimental Design Limitations

Lack of Long-Term Exposure Models:The study uses short-term MS exposure (4 days) followed by acute infection, which does not mimic clinical moxibustion practices (often involving repeated sessions). Chronic toxicity endpoints (e.g., lung histology after 28 days, systemic organ damage) are absent, precluding assessment of cumulative risks.

Inadequate Control of Confounding Variables:The study does not specify the combustion conditions of moxa sticks (e.g., temperature, airflow), which profoundly influence smoke composition. Variations in moxa stick quality (e.g., age of Artemisia argyi, processing methods) may lead to inconsistent results, affecting reproducibility.

Recommendations for Revision

1. Expand Safety Evaluation:

Include chronic toxicity experiments in mice (e.g., 4-week MS exposure without infection) to assess pulmonary fibrosis (e.g., collagen deposition via Masson’s trichrome staining), MMP12 activity, and systemic organ toxicity (liver/kidney function tests).

Characterize MS components using advanced analytical techniques (e.g., GC-MS, HPLC) to identify potential carcinogens/endocrine disruptors, and reference occupational exposure limits (e.g., OSHA standards for formaldehyde, PM2.5).

2. Clarify Mechanistic Paradoxes:

Investigate the temporal and dose-dependent effects of MS on MMP12 expression. Is MMP12 upregulation a late-phase response or specific to high concentrations?

Perform gain/loss-of-function experiments (e.g., PPARγ agonists/antagonists) to dissect its dual role in inflammation and viral clearance.

3. Enhance Clinical Translational Data:

Conduct a retrospective survey of acupuncturists’ respiratory health to correlate MS exposure with chronic pharyngitis or COPD incidence.

Develop low-toxicity MS formulations (e.g., smokeless moxibustion) and compare their efficacy/toxicity with traditional MS in parallel experiments.

4. Revise Discussion Section:

Acknowledge the trade-off between MS’s therapeutic benefits and safety risks, particularly for vulnerable populations (pregnant women, elderly, individuals with preexisting lung disease).

Highlight the need for standardized MS protocols in clinical settings to minimize exposure while preserving efficacy.

Reviewer #2: This manuscript is valuable for Investigation of the Effect and Mechanisms of Moxa Smoke in the Treatment of Influenza A Virus (IAV) Infection. But there are some important data need to clarify.

1. In page 9, line 182-186. The moxa smoker exposure experiment is low moxa smoke (LMS, resulting in a PM10 concentration of approximately 1.335 mg/m³) and high moxa smoke (HMS, resulting in a PM10 concentration of approximately 2.67 mg/m³)21. Both PM10 concentrations were lower than the typical concentration of 3.54 mg/m³ found in moxibustion clinics. But the dosage conversion isn’t correct simply by same unit (mg/m³) for animal and human. As common view, we usually use body surface area method to calculate drug dosage between different species. So in Figure 5, we could find the body weight loss of LMS+PR8 group and HMS+PR8 group reduced seriously compared MEM+sham group from day 2-4 before H1N1 PR8 infection. The author needs to explain these moxa smoke exposure concentrations and also need more data to prove its safety in this dosage.

2. In page 23, H1N1 NP gene of MEM+sham group is not correct, it is normal mice, it’s impossible to have H1N1 NP gene. So, these data need to calculate again.

3. Did author analysis the number of inflammatory cells in BALF with flow cytometry? If this, the author need provide more experimental information with antibody, company name, experimental condition etc.

4. Author need to check style of reference, English expression carefully in this manuscript.

5. The image resolution is insufficient, need to improve.

**Do you want your identity to be public for this peer review?** For information about this choice, including consent withdrawal, please see our Privacy Policy

Reviewer #1: No

Reviewer #2: No

---

## [Author Response · Author response to Decision Letter 1]

2 Nov 2025

Dear editor and reviewers:

We thank all the editors and reviewers for their valuable comments and suggestions. We have carefully revised the manuscript to enhance its clarity and·facilitate the understanding of the readers. Our point-to-point responses are presented in the following." We hope that the revision would satisfactorily" address the comments and conceits of the editors and reviewers.

Below, we have addressed all the additional requirements as outlined in your feedback:

Comments from the Editor:

Response We have carefully revised the manuscript to comply with PLOS ONE’s style requirements, using the provided templates. All files have been renamed according to the journal’s guidelines.

Response Our research doesn’t involve any author-generated code.

“Guangzhou Key Laboratory of Traditional Chinese Medicine for the Prevention and Treatment of Chronic Cough and Dyspnea (Grant Number 2023A03J0226), the double world-class and high-level university discipline collaborative innovation team project of Guangzhou University of Chinese Medicine (Grant Number 2021XK27).”

Response: We have updated the information regarding fund allocations. This update includes the addition of the fund named ‘Investigation into the Optimal Dosage of Flu Dual-Relief Granules for the Treatment of Viral Pneumonia and the Underlying Mechanisms Involving Energy Metabolism Regulation in Macrophage Polarization (Grant Number 2022B1515230001)’, which covers publication expenses. We have also clarified the role of all the funds, as per your request, in both the manuscript and the cover letter.

4. We note that your Data Availability Statement is currently as follows: All relevant data are within the manuscript and in Supporting Information files.

Response: We confirm our submission contains the complete minimal data set required to replicate all study findings. It has been uploaded as Supporting Information files [File names: S2_raw_datasets]

5. Please include a copy of Table 4 which you refer to in your text on page 22. On page 22,

Response: "Table 4" was erroneously noted - this has been corrected to "Table 3" throughout the manuscript. The complete Table 3 is now properly included in the main document on page 24.

Response All original, uncropped blot/gel images have been uploaded as Supporting Information files [File names: S1_raw_images]

Comments from reviewers

Reviewer #1

Overall Evaluation

This study explores the antiviral and anti-inflammatory effects of moxa smoke (MS) against H1N1 infection using network pharmacology, molecular docking, and in vivo experiments. The integration of traditional Chinese medicine (TCM) with modern pharmacological approaches is commendable, offering novel insights into MS’s potential therapeutic applications. However, while the research demonstrates scientific rigor in certain aspects, it contains critical gaps, particularly regarding clinical safety implications and mechanistic clarity, which must be addressed for publication.

Strengths and Innovations

1. Multidisciplinary Approach to Mechanism DiscoveryThe study employs network pharmacology to systematically identify MS components (52 compounds) and their potential targets (384 proteins), uncovering 16 core targets (e.g., STAT3, PPARγ) via PPI network analysis. This “multi-component, multi-target” framework aligns with TCM’s holistic philosophy and provides a robust basis for understanding MS’s complex actions. The molecular docking validation of stable binding between MS compounds (e.g., Bis-(3,5,5-trimethylhexyl) phthalate) and key targets further strengthens the mechanistic hypothesis.

2. In Vivo Validation of Antiviral and Anti-Inflammatory EffectsUsing a PR8-infected mouse model, the study demonstrates that MS reduces viral load (H1N1NP mRNA), alleviates pulmonary inflammation (e.g., decreased BALF leukocytes, IL-6/TNF-α expression), and modulates PPARγ/STAT3 pathways. These findings provide experimental evidence for MS’s therapeutic potential, particularly its ability to suppress cytokine storms—a critical pathology in severe influenza.

3. Translational Relevance to TCM PracticeBy focusing on MS, a byproduct of moxibustion with historical use in infection prevention, the study bridges traditional practices with modern science. The identification of PPARγ as a key regulator offers a mechanistic explanation for MS’s anti-inflammatory effects, potentially guiding the development of targeted TCM interventions.

Critical Weaknesses and Safety Concerns

1. Inadequate Assessment of Clinical Safety Risks

Occupational Exposure Risks for Acupuncturists: The manuscript fails to address the chronic toxicity of MS, a major concern for acupuncturist) and patients. Existing evidence (cited in prior reviews) shows that long-term exposure to moxa smoke is associated with increased incidence of chronic pharyngitis, allergic reactions, and potential pulmonary fibrosis. For example, elevated MMP12 expression in MS-treated mice (Table 3) indicates ECM degradation, a hallmark of emphysema and pulmonary fibrosis. The study’s failure to discuss this finding’s clinical relevance constitutes a significant oversight.

Systemic Toxicity of Smoke Components: MS contains known toxicants, including PM2.5/PM10, aldehydes (e.g., formaldehyde), and polycyclic aromatic hydrocarbons (PAHs). While the study excludes “toxic gaseous compounds,” it provides no data on the safety of identified components (e.g., phthalates, terpenoids) in long-term use. For instance, phthalates are endocrine disruptors linked to metabolic and reproductive disorders, and PAHs are carcinogenic. The absence of toxicity profiling (e.g., hepatic/renal function assays, genotoxicity tests) undermines the safety narrative.

Response We appreciate the reviewers' comments. Indeed, prolonged exposure to moxa smoke poses significant occupational hazards for acupuncturists that warrant careful attention. Through a comprehensive literature review and our team's qualitative LC-MS analysis of moxa smoke, we found the compounds such as benzaldehyde, phenol, benzonitrile, and other toxic substances known to cause respiratory irritation, hepatotoxicity, nephrotoxicity, metabolic disruption, and even carcinogenic effects. Additionally, inhalable particulate matter PM10 was detected. In response to the reviewers’ comments, we further investigated the effects of both short-term and long-term moxa smoke exposure on hepatic and renal function, organ mass, pulmonary tissue damage, and collagen deposition. In conjunction with the results of previous clinical studies, it thoroughly evaluates the safety of short-term exposure and the toxicity associated with long-term exposure, as well as the underlying material basis.

These refinements enhance the safety considerations and clinical relevance of the findings, making the research more comprehensive and systematic. Specific results are detailed in the Recommendations for Revision (Page 9-16). The reviewers' suggestion to conduct research on the long-term toxicity of harmful substances in smoke is a valuable recommendation, and we shall progressively undertake such studies in our future work.

2. Mechanistic Gaps and Incomplete Data Interpretation

Contradictory Findings with MMP12 Upregulation: The study reports that high-dose MS (HMS) significantly upregulates MMP12 mRNA in lung tissue (p<0.05), yet dismisses this as a “potential risk” without mechanistic exploration. MMP12’s role in alveolar wall destruction and fibrosis is well-established; its induction by MS contradicts the claimed “protective effect” on lung injury. The authors must clarify whether this is a dose-dependent effect or an experimental artifact, ideally through immunohistochemistry or functional assays (e.g., ECM degradation markers).

Unclear Dynamics of PPARγ/STAT3 Pathways: While MS upregulates PPARγ and reduces p-STAT3, the study does not address how these changes balance antiviral immunity and inflammatory suppression. For example, PPARγ overactivation may impair macrophage antiviral function, while STAT3 inhibition could compromise viral clearance. Long-term outcomes (e.g., viral persistence, secondary infections) are not evaluated, limiting conclusions about therapeutic windows.

Response� We are grateful for the reviewer’s insightful questions. Elevated MMP12 expression is indeed a significant contributing factor in the development of pulmonary emphysema and fibrosis. However, the research team maintains that this does not contradict the role of MS in suppressing viral replication and mitigating inflammatory storms. Rather, it emphasises the necessity of thoroughly considering its safety profile in clinical applications. This can be achieved through modern scientific techniques, such as purifying the active constituents and removing particulate matter and toxic substances, thereby minimising potential adverse effects. This study, employing network pharmacology methods alongside toxicological research, is well suited to advancing the identification of both therapeutic and toxic components, thereby facilitating further development. In response to the reviewer’s suggestions, this study examined the dose- and time-dependent effects of moxa smoke on MMP12. Combining pulmonary pathology and collagen detection results, we discuss the reasons and correlations between elevated MMP12 levels induced by both short-term and long-term (four weeks) MS exposure, and pulmonary tissue destruction. Relative results and discussion are detailed in the Recommendations for Revision (Page16).

Regarding the PPARγ/STAT3 pathway, we further validated the efficacy of high-dose MS against PR8 viral infection. Additionally, we investigated the effects of PPARγ agonists�and the combined effect of high-dose MS with PPARγ inhibitors. This approach explored the anti-inflammatory mechanism of MS, which depends on PPARγ activation. We discovered that MS exhibits antiviral activity independent of PPARγ, revealing its unique pharmacological property of achieving a balanced effect between anti-inflammatory and antiviral actions despite apparent contradictions. Specific results are detailed in the Recommendations for Revision (Page17-20).

This supplementary data further facilitates a systematic elucidation of the effects of MS on tissue destruction and collagen deposition, clarifies its anti-inflammatory mechanisms, and enhances the depth of the research.

3. Experimental Design Limitations

Lack of Long-Term Exposure Models: The study uses short-term MS exposure (4 days) followed by acute infection, which does not mimic clinical moxibustion practices (often involving repeated sessions). Chronic toxicity endpoints (e.g., lung histology after 28 days, systemic organ damage) are absent, precluding assessment of cumulative risks.

Inadequate Control of Confounding Variables:

The study does not specify the combustion conditions of moxa sticks (e.g., temperature, airflow), which profoundly influence smoke composition. Variations in moxa stick quality (e.g., age of Artemisia argyi, processing methods) may lead to inconsistent results, affecting reproducibility.

Response� We acknowledge the shortcomings of this study as highlighted by the reviewers. Indeed, the experimental design employed short-term MS exposure (4 days) followed by acute infection, rather than post-infection treatment, which does not fully replicate clinical moxibustion practice. This limitation primarily arises from constraints within the experimental conditions. Our smoke exposure protocol required placing mice from the same treatment group together in a single smoke-exposure chamber three times daily, with the chamber lid opened every 25 minutes within a fume cupboard to ensure ventilation. Administering MS exposure after viral challenge would have significantly increased the risk of aerosol transmission of the virus among groups, thereby compromising the reliability of the experimental results. Consequently, we adopted the approach of administering treatment prior to viral challenge. Fortunately, the beneficial anti-inflammatory and antiviral effects of moxa smoke were observed after only four days of exposure.

Influenza virus infection is inherently a self-limiting disease. Following infection, mice reach peak viral replication and inflammation within 3–5 days, with the virus completely cleared around day 7-10. This pattern mirrors that of human influenza infection. Consequently, our therapeutic strategy prioritises achieving effectiv

---

## [Decision Letter · Decision Letter 1]

17 Nov 2025

Investigation of the Effect and Mechanisms of Moxa Smoke in the Treatment of Influenza A Virus (IAV) Infection

PONE-D-25-21289R1

Dear Dr. Yu,

We’re pleased to inform you that your manuscript has been judged scientifically suitable for publication and will be formally accepted for publication once it meets all outstanding technical requirements.

Kind regards,

Md Bashir Uddin, PhD

Academic Editor

PLOS ONE

Additional Editor Comments (optional):

Reviewers' comments:

Reviewer's Responses to Questions

**Comments to the Author**

Reviewer #3: All comments have been addressed

2. Is the manuscript technically sound, and do the data support the conclusions?

Reviewer #3: Partly

3. Has the statistical analysis been performed appropriately and rigorously?

Reviewer #3: Yes

4. Have the authors made all data underlying the findings in their manuscript fully available?

Reviewer #3: Yes

5. Is the manuscript presented in an intelligible fashion and written in standard English?

Reviewer #3: Yes

Reviewer #3: Concerns raised in the previous revision round have been satisfactorily addressed. Although there are limitations in the experimental design, the manuscript has benen improved based on the previous comments

**Do you want your identity to be public for this peer review?** For information about this choice, including consent withdrawal, please see our Privacy Policy

Reviewer #3: No

---

## [Editor Report · Acceptance letter]

PONE-D-25-21289R1

PLOS ONE

Dear Dr. Yu,

I'm pleased to inform you that your manuscript has been deemed suitable for publication in PLOS ONE. Congratulations! Your manuscript is now being handed over to our production team.

Kind regards,

on behalf of

Dr. Md Bashir Uddin

Academic Editor

PLOS ONE